# Microporous polymer adsorptive membranes with high processing capacity for molecular separation

Zhenggong Wang[1], Xiaofan Luo[2], Zejun Song[2], Kuan Lu[3], Shouwen Zhu[1], Yanshao Yang[2], Yatao Zhang [4], Wangxi Fang [2✉] & Jian Jin [1✉]

Trade-off between permeability and nanometer-level selectivity is an inherent shortcoming of membrane-based separation of molecules, while most highly porous materials with high adsorption capacity lack solution processability and stability for achieving adsorption-based molecule separation. We hereby report a hydrophilic amidoxime modified polymer of intrinsic microporosity (AOPIM-1) as a membrane adsorption material to selectively adsorb and separate small organic molecules from water with ultrahigh processing capacity. The membrane adsorption capacity for Rhodamine B reaches 26.114 g m$^{-2}$, 10–1000 times higher than previously reported adsorptive membranes. Meanwhile, the membrane achieves >99.9% removal of various nano-sized organic molecules with water flux 2 orders of magnitude higher than typical pressure-driven membranes of similar rejections. This work confirms the feasibility of microporous polymers for membrane adsorption with high capacity, and provides the possibility of adsorptive membranes for molecular separation.

[1] Innovation Center for Chemical Science, College of Chemistry, Chemical Engineering and Materials Science & Collaborative Innovation Center of Suzhou Nano Science and Technology, Soochow University, Suzhou, China. [2] i-Lab, Suzhou Institute of Nano-Tech and Nano-Bionics, Chinese Academy of Sciences, Suzhou, China. [3] State Key Laboratory of Coal Conversion, Institute of Coal Chemistry, Chinese Academy of Sciences, Shanxi, China. [4] School of Chemical Engineering and Energy, Zhengzhou University, Zhengzhou, China. ✉email: wxfang2017@sinano.ac.cn; jjin@suda.edu.cn

Molecular separation is an essential component in many human daily activities and multiple industrial, medical, and environmental processes, such as water purification, oil and gas refining, energy generation and storage, pharmaceutical ingredient extraction and purification[1–5]. Among various materials and methods, synthetic membranes and membrane-based separation have been extensively studied due to its strong sustainability, scale-up feasibility, and no phase change during separation[1,6]. However, trade-off between membrane permeability and selectivity is an inherent shortcoming of membrane-based molecular separation[7–10]. Taking pressure-driven membrane processes for instance, microfiltration and ultrafiltration membranes exhibit high water permeation flux and reject large molecules with high efficiency, but their large pores (0.002–1 μm) are unable to separate nanometer-sized small organic molecules[11]. Processing of these small organic molecules requires membranes of nanofiltration level pore size and selectivity, yet the reduction in pore size inevitably results in much lower water permeation flux[12]. In contrast, adsorption can be a highly selective molecular separation process with specific physical or chemical interactions between adsorbents and target molecules[13–16], but its application is often limited by its low processing rate, high internal diffusion resistance within absorbents, etc[17,18]. Membrane adsorption is a pressure-driven dynamic membrane-based adsorption process, which combines the merits of both adsorption and membrane separation[19–21]. Compared to traditional pressure-driven membranes with size sieving as the dominant separation mechanism, adsorptive membranes utilize more specific membrane-solute interactions like electrostatic interactions, π–π interactions, van der Waals forces, and hydrogen bonding to achieve highly selective and fast separation of small organic molecules[22,23]. Therefore, they are expected to break through the permeability-selectivity trade-off by simultaneously achieving the selectivity of dense membranes and permeability of porous membranes[20,24–27].

However, the application of membrane adsorption is hindered by the lack of adsorptive membranes with sufficient processing capacity, which greatly affects their further development and practical application. Adsorptive membranes are usually prepared by post-grafting affinity ligands or adsorptive filler mixing in traditional polymer materials[17,22]. Due to the limited specific surface area and adsorption sites incorporated, the overall processing capacity of these conventional adsorptive membranes is very low, mostly in the range of 0.01–1 g m$^{-2}$. Such membranes could only handle solutions of very low concentration and require frequent cleaning and regeneration, which greatly affects their further development and practical application. Although porous materials such as metal-organic frameworks (MOFs) possess large porosities and rich affinity sites[20,21,28–33], their poor solution processability hinders their engagement as absorptive membrane separation materials. Besides, most MOFs materials do not have sufficient stability in water, especially under acidic or alkaline conditions, resulting in membrane failure during long-term operation[8].

In this work, we report a microporous polymer based adsorptive separation membrane, which has very high adsorption capacity and can realize fast and selective molecular separation. The microporous polymer is amidoxime modified polymer of intrinsic microporosity (AOPIM-1) polymer, which owns a rigid and contorted three-dimensional structure in its backbone. The ineffective chain packing brings high specific surface area up to 550 m$^2$ g$^{-1}$ and produces interconnected free volume elements with a size of less than 2 nm[34–38]. The amidoxime modification endows the polymer with good solution processability and provides abundant adsorption sites for selectively adsorbing charged molecules. Due to its unique chemical and physical structure, it achieves a high-efficiency removal of small organic molecules (>99.9%) with permeating flux 2 orders of magnitude higher than typical nanofiltration membranes of similar dye rejections, and such separation performance is maintained throughout multiple adsorption-elution cycles with >98% flux recovery rate (FRR). More importantly, the static adsorption capacity of the membrane material greatly surpasses traditional non-porous polymer adsorbents and is comparable to MOF-based adsorbents, while the dynamic processing capacity reaches 26.114 g m$^{-2}$, much higher than all the adsorptive membranes reported so far. The high processing capacity is expected to enable the membrane towards more realistic adsorption-separation application scenarios.

## Results and discussion

**Material characterization of AOPIM-1 polymer.** Polymer of intrinsic microporosity (PIM-1) was synthesized via a polycondensation reaction between TTSBI and TFTPN (Supplementary Fig. 1) as reported previously[39], which shows a high specific surface area (786 m$^2$ g$^{-1}$). The prepared PIM-1 was further modified with hydroxylamine to obtain AOPIM-1 (Fig. 1a, Supplementary Fig. 2).[34,37] Spectroscopic characterizations of AOPIM-1 are provided in Supplementary Figs. 3, 4, which signify the successful synthesis of the polymer. Figure 1d shows a three-dimensional view of a modeled amorphous cell of the AOPIM-1 polymer, which presents a highly microporous feature. Its specific surface area reaches 552 m$^2$ g$^{-1}$ as deduced from its N$_2$ adsorption isotherm (Fig. 2a). Moreover, the amidoxime modified polymer changes its solubility parameter due to the introduction of polar groups, making it soluble in common casting solvents, such as DMF, NMP, DMSO, etc (Supplementary Fig. 5). Unlike PIM-1 that could only be dissolved in chloroform and tetrahydrofuran, the great solution processability of AOPIM-1 makes it feasible for the fabrication of asymmetric membranes via the industrially scalable phase inversion approach (Fig. 1a). In addition, stability of AOPIM-1 under acidic and alkaline conditions is examined by immersing in acid (pH = 3) and base (pH = 10) for at least 24 h, and the chemical and physical structure of the polymer is found to remain unchanged (Fig. 2a, Supplementary Fig. 4). All the membranes in various pH condition demonstrate sufficient mechanical strength for pressurized permeation tests (Supplementary Fig. 6), while the mechanical strength appears lower in acid condition, likely due to decreased interchain H-bond interaction. Interestingly, the AOPIM-1 polymer shows a pH-tunable chargeability (Fig. 1b). Such tunable chargeability is attributed to the protonation/deprotonation of amidoxime groups, and they could thus be utilized as effective affinity sites with pH-tunability for selective adsorption and separation of oppositely charged molecules (Fig. 1c). The processability, hydrophilicity, high specific surface area and pH-tunable affinity sites make AOPIM-1 a promising membrane adsorption material for aqueous-based molecular separations, which will be demonstrated in the following sections.

**pH-tunable static adsorption feature of AOPIM-1 polymer.** Two organic dye molecules with similar molecular weight but different charge properties, Methyl Orange (MO, MW = 327, negatively charged) and Methylene Blue (MB, MW = 320, positively charged) are used to evaluate the static adsorption behavior of AOPIM-1. As shown in Fig. 2b, AOPIM-1 favors the capture of negatively charged MO molecules under acidic conditions while capturing the oppositely charged MB under alkaline conditions with the equilibrium adsorption capacity higher than 450 mg g$^{-1}$. And the capacity falls below 100 mg g$^{-1}$ under neutral pH conditions where the amidoxime group possess minimal

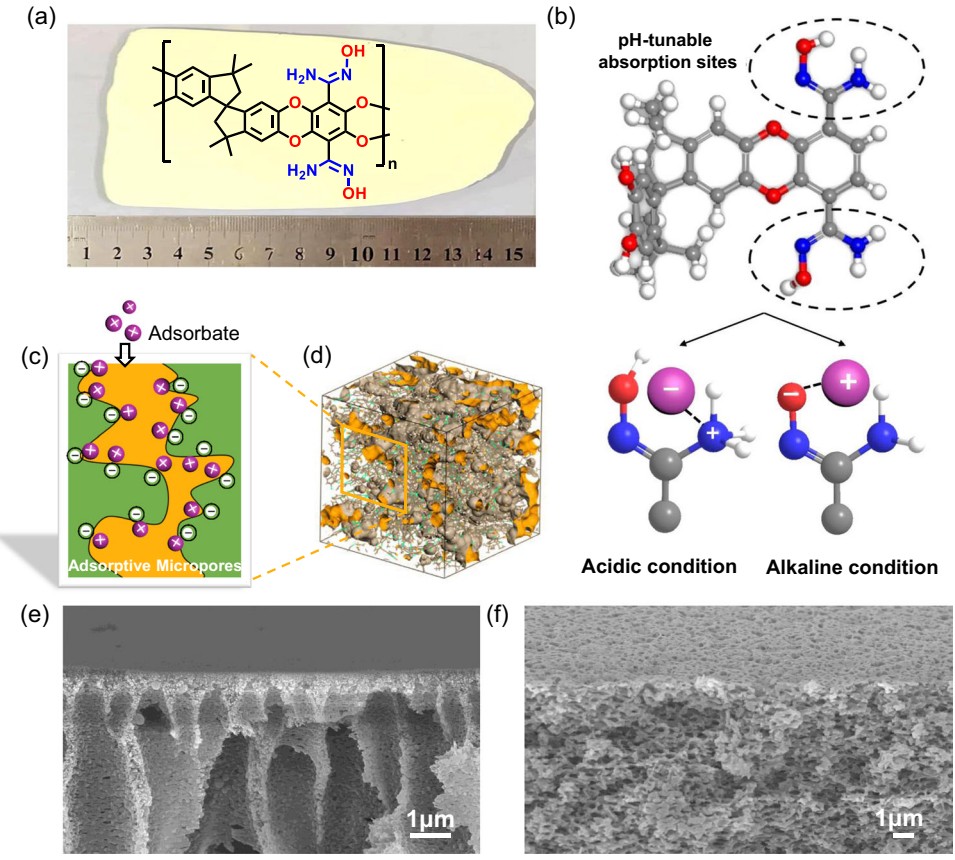

**Fig. 1 Fabrication of asymmetric AOPIM-1 membranes. a** Chemical structure of AOPIM-1 polymer and optical image of AOPIM-1 membrane; **b** The pH-tunable chargeability of AOPIM-1 under acidic or alkaline conditions, where the gray, white and red spheres represent C atoms, H atoms and O atoms, respectively; **c** Schematic diagram of microporous polymer membranes for adsorptive separation of organic molecules; **d** Three-dimensional view of an amorphous cell of the AOPIM-1 polymer. The brown surface indicates the van der Waals surface, and the orange surface is the Connolly surface with a probe radius of 1.6 Å; Cross-section SEM images of the AOPIM-1 membranes obtained by phase conversion of different coagulation bath compositions: **e** $H_2O$:EtOH = 100:0 and **f** $H_2O$:EtOH = 0:100.

chargeability, which reveals that the pH-tunable affinity sites make major contribution to the adsorption capacity. Therefore, the dye molecules separation by AOPIM-1 membrane should be mainly attributed to electrostatic interactions between charged amidoxime groups and oppositely charged dye molecules. Bekir et al.[40] also reported the adsorption of charged dyes, methylene blue (MB) and methyl orange (MO), from water system by AOPIM-1 powder. The experimental adsorption capacities of AOPIM-1 are 79.8 mg/g and 69.8 mg/g for MO and MB at pH 6, respectively, which is consistent with our data. As shown in Supplementary Fig. 7, the mixture of MB and MO is green with some precipitation due to the opposite charges of MB and MO molecules[16]. After AOPIM-1 treatment under different pH conditions, one of the colors is completely removed, and the precipitation disappears. The UV-Vis spectra clearly show the disappearance of one of the absorption peaks. The result proves that AOPIM-1 achieves the selective adsorption of MB and MO mixture by adjusting pH environment. Energy optimized molecular model and adsorption energies of AOPIM-1 to MB, MO and $H_2O$ molecule in alkaline condition are shown in Supplementary Fig. 8. The absorption energy ($E_{ads}$) of AOPIM-1 to MB, MO and $H_2O$ molecule are −1587.7, −745.4, and −742.3 kcal/mol, respectively. Obviously, the absorption energy ($E_{ads}$) of AOPIM-1 to MB is the highest, which is very consistent with the experimental result that MB molecule was selectively adsorbed and separated by the AOPIM-1 membrane. In Fig. 2c, d, Langmuir adsorption isotherms and pseudo-second-order equation

($R^2 > 0.99$) are used to fit the adsorption of AOPIM-1 on target molecules in different pH environments. Under acidic and alkaline conditions, the maximum adsorption capacity for MO and MB reaches 491.63 mg g$^{-1}$ and 765.09 mg g$^{-1}$, respectively. AOPIM-1 shows a faster adsorption rate, exceeding 50% in 40 min, and reaching equilibrium in 300 min. The obtained adsorption capacity of AOPIM-1 is much higher than traditional non-porous polymer adsorbents and comparable to that of newly developed MOF adsorbents (Supplementary Tab. 1)[20,33,39–46].

**Adsorption separation of dye molecules by AOPIM-1 membranes**. The adsorptive membranes based on AOPIM-1 are constructed via a wet-phase inversion method. The membrane structure is altered through tuning phase inversion conditions including various coagulation bath composition, casting solution concentration, and casting solution composition as summarized in Supplementary Tab. 2. With increasing ethanol content in coagulation bath, it could be observed that the structure of the membrane changes from finger-like pores (M1) to sponge-like pores (M3) as demonstrated in the cross-sectional SEM images in Fig. 1e, f. Meanwhile, the pure water flux of the resulting membrane decreases from 1505.7 L m$^{-2}$ h$^{-1}$ bar$^{-1}$ (M1) to 249.8 L m$^{-2}$ h$^{-1}$ bar$^{-1}$ (M3), which is consistent with the decrease trend of pore size distribution (Supplementary Fig. 9). However, the processing capacity of M3 increases from 3.7 to 11.4 g m$^{-2}$ which is 3 times more than that of M1 (Supplementary Tab. 2, Supplementary Fig. 10-12). It should be

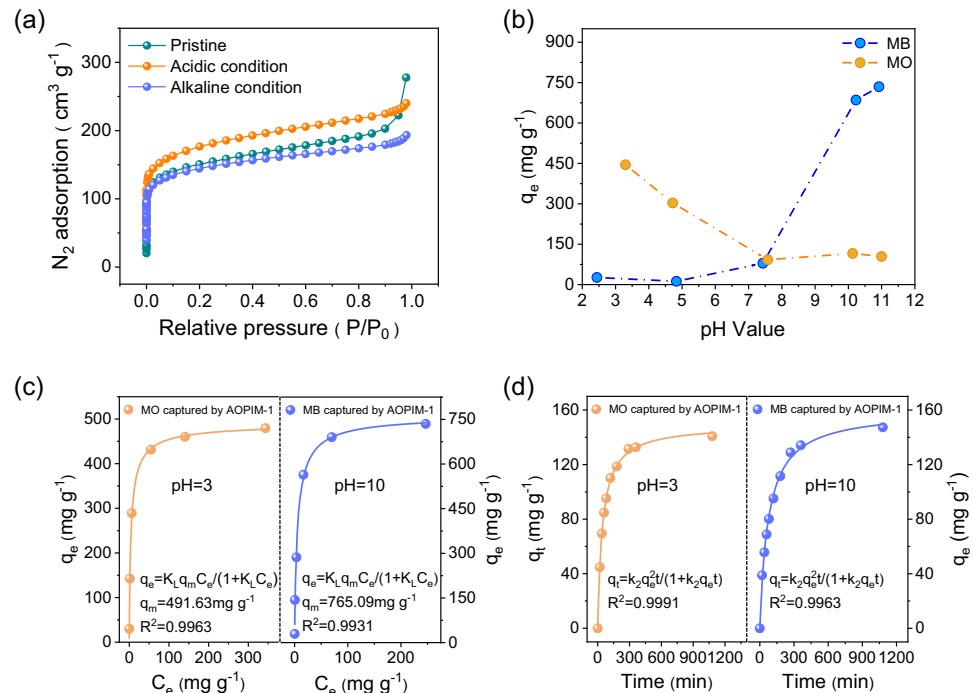

**Fig. 2 pH-tunable static adsorption feature of AOPIM-1 polymer. a** Nitrogen absorption isotherms of AOPIM-1 as-prepared and after immersing in acid (pH = 3) or base (pH = 10) for at least 24 h; **b** Equilibrium adsorption capacity ($q_e$) of AOPIM-1 for MO and MB dye molecules at different pH conditions; **c** Adsorption isotherm of MO and MB by AOPIM-1 at acidic and alkaline conditions, respectively ($q_e$: equilibrium adsorption capacity, $q_m$: maximum adsorption capacity, $C_e$: concentration of dye molecules at equilibrium); **d** Adsorption of MO and MB by AOPIM-1 as a function of contact time at acidic and alkaline conditions, respectively ($q_t$: adsorption amount at time $t$).

mentioned that the effective processing capacity of membranes is calibrated under the criterion of 99% rejection ratio of Rhodamine B (RHB) (20 ppm). The solutes concentration influence on the membrane separation performance could be found in the supporting information (Supplementary Fig. 13). With increase of the concentration of the casting solution, it could be observed that the membrane thickness (M3-M5 in Supplementary Tab. 2) is gradually increased. The increase in thickness and processing capacity has a linear relationship from M3 11.4 g m$^{-2}$ (64 μm) to M5 26.1 g m$^{-2}$ (119 μm). Figure 3e and Supplementary Tab. 3 present the comparison of AOPIM-1 membranes with reported adsorptive membranes with respect to their permeation flux and adsorptive capacity, and the dynamic adsorption capacity of the membrane appears to be 10–1000 times higher than previously reported adsorptive membranes. We also calculated processing capacity of the AOPIM-1 membranes in the unit of g/m$^3$, which is 28, 45, 178, 191 and 219 kg/m$^3$ for M1, M2, M3, M4, and M5, respectively. Interestingly, the volumetric processing capacity does not only increase sharply when the membrane changes from finger-like structure to sponge-like structure (M1 to M3), but also found to increase with the membrane thickness (M3 to M5). It appears that the membrane processing capacity is closely related to the length of dye transportation route on top of pore volume. The longer and tortuous route would result in higher processing capacity, which is consistent with the property of adsorption materials. The above experiment results prove that by adjusting the phase inversion process, the membrane structure can be reasonably designed to achieve the purpose of rapidly removing small molecular organic pollutants in aqueous system using dynamic adsorption processes. To provide direct evidence of the adsorption and distribution of molecules in the membranes, SEM characterization of "saturated" AOPIM-1 membrane (the membrane at the point where the dye rejection falls below 99% during the dynamic adsorption experiment) is

conducted. Methylene blue (MB) is selected as adsorbate because MB contains sulfur elements that can be determined by EDX mapping. The membrane is dried via freeze-drying to avoid possible structural feature collapse. The obtained membrane is characterized by SEM imaging and EDX mapping. The surface and cross-sectional SEM and EDX mapping images are shown in Supplementary Fig. 14. There is no obvious cake-layer found in the membrane surface. Instead, it could be clearly seen that the sulfur element is widely distributed across the whole cross-section of the membrane, especially among the polymer-rich regions. This result provides direct evidence of the adsorption and distribution of molecules in the AOPIM-1 membrane. It also demonstrates directly that the dye molecules are adsorbed in the polymer micropores across the entire membrane until saturation, and then penetrates through the membrane afterwards.

The dynamic adsorption separation process was conducted on a dead-end setup using a sponge-like AOPIM-1 membrane (M3). According to zeta potential measurement results, AOPIM-1 membranes exhibit positive charge at acidic pH < 4 while turning into negative charge at alkaline aqueous environment (Fig. 3a). Thus, six types of organic dyes with different sizes and charges were used as target molecules for separation. Negatively charged dyes including Methyl Orange (MO), Congo Red (CR) and (BB) were tested in acidic condition and positive charged dyes including Methylene Blue (MB), RHB and Crystal Violet (CV) (Supplementary Fig. 15) was tested in alkaline condition. It can be seen from Fig. 3b and Supplementary Tab. 4 that under acidic conditions (pH = 3), MO (99.9 % rejection, 203.8 L m$^{-2}$ h$^{-1}$ bar$^{-1}$), CR (99.9 % rejection, 180.9 L m$^{-2}$ h$^{-1}$ bar$^{-1}$) and BB (99.9% rejection, 192.5 L m$^{-2}$ h$^{-1}$ bar$^{-1}$) can be selectively retained. Under alkaline conditions (pH = 10), MB (99.9 % rejection, 177.9 L m$^{-2}$ h$^{-1}$ bar$^{-1}$), RHB (99.9% rejection, 191.3 L m$^{-2}$ h$^{-1}$ bar$^{-1}$) and CV (99.9 % rejection, 170.0 L m$^{-2}$ h$^{-1}$ bar$^{-1}$) can be retained, indicating highly efficient

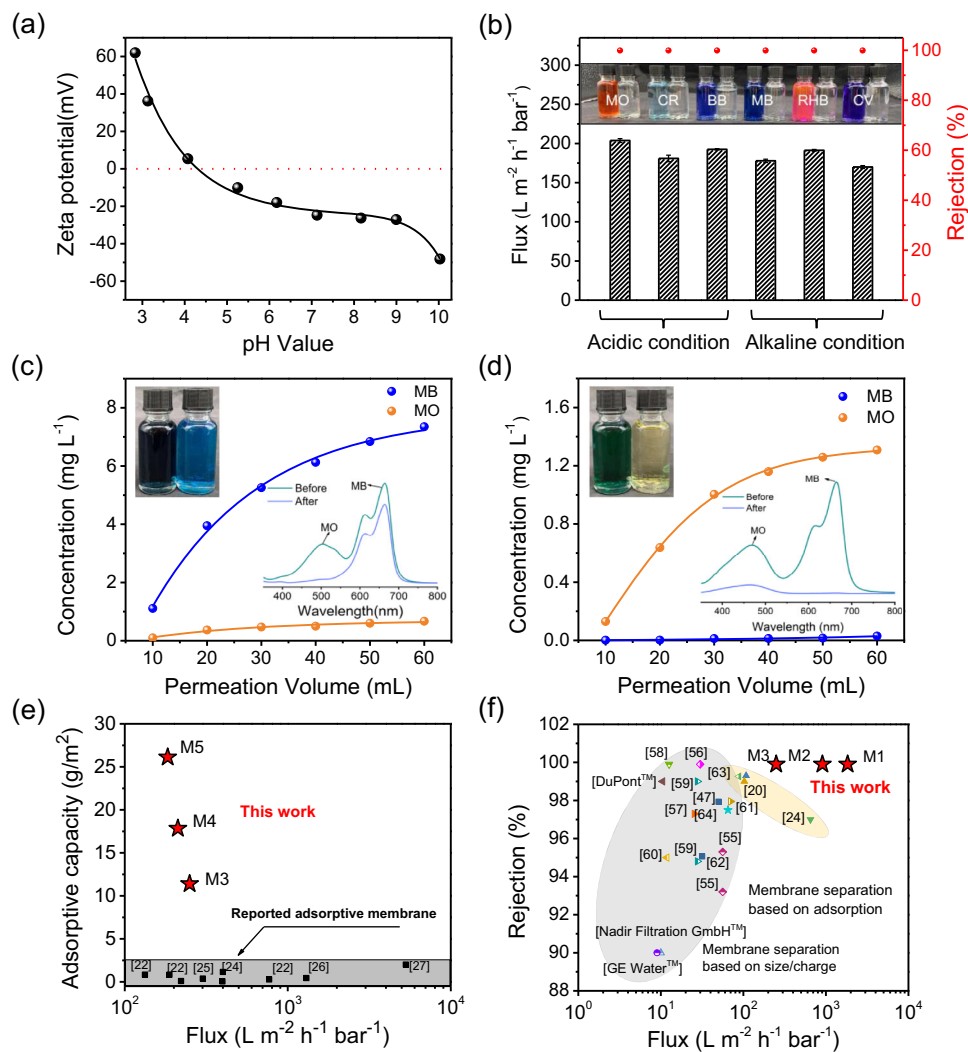

**Fig. 3 Charge-selective dynamic adsorption of organic dye molecules by AOPIM-1 membranes. a** Zeta potential of AOPIM-1 membrane at pH 3-10; **b** The dynamic separation performance of AOPIM-1 membrane at different pH condition for different dyes (concentration 20 ppm, applied pressure 0.2 MPa, pH = 3/10); **c, d** The separation (MO/MB) of two dye molecules with similar molecular weights but different charges under different pH condition (concentration 10 ppm MO + 10 ppm MB, applied pressure 0.2 MPa, pH = 3/10); **e** Comparison of AOPIM-1 membranes with reported adsorptive membranes with respect to their permeation flux and adsorptive capacity[22,24–27]; **f** Comparison of AOPIM-1 membranes with reported membranes with respect to their permeation flux and dye rejection/removal efficiency. The effective flux and processing capacity of the membrane in this study is calibrated under the criterion of 99% rejection ratio of dye molecule[20,24,47,55-64].

separation of dye molecules with different charges. As shown in Supplementary Fig. 16, the membrane adsorption capacity in neutral condition is only 0.3 and 7.1 g/m² for MO and RHB, respectively, much lower than that in acid or alkaline condition. This is ascribed to the minimal chargeability in neutral condition. It also illustrates that the pH-tunable affinity sites make dominant contribution to the adsorption capacity of our membrane. The dynamic separation of two dye molecules mixture (MO/MB) is further demonstrated (Fig. 3c, d). Under different pH conditions, one of the colors of the mixed solution disappears while the other color remains after filtering. The UV–vis spectra clearly show the disappearance of one of the absorption peaks. The results prove that the AOPIM-1 membrane has the characteristic of charge selective dynamic adsorption of dye molecules. We summarized the state-of-the-art nanofiltration membranes and other membrane adsorption materials reported in the literature which are utilized for separating small organic molecules in aqueous systems (Fig. 3f), and the current AOPIM-1 membranes possess 2 orders of magnitude higher flux than typical nanofiltration membranes of similar dye rejections, and

surpass other reported adsorptive separation membranes in terms of water flux and dye rejection.

The adsorptive separation behavior of the AOPIM-1 membrane is further demonstrated in a multi-cycle dynamic membrane adsorption process. It can be seen from Fig. 4a that the flux of the membrane decreases with the increasing permeation volume in each cycle, which can be attributed to the accumulation of dyes on the surface and inside of the adsorptive membrane. In the first seven cycles, 20–30 mL of methanol was adopted as a feed solution to desorb the dye molecules from the membrane for a short time (within 5 min). The concentration of feed and permeate in each cycle is summarized in Supplementary Tab. 5. It can be seen in digital photo in Fig. 4d that the color of the membrane due to the adsorption of dye disappears after cleaning. At the same time, the flux recovery rate (FRR) in each cycle reaches 98%. After 7 cycles, a long-term desorption treatment (about 2 hours by methanol) on the membrane was performed, and a higher recovery rate is got. The Brunauer-Emmet-Teller (BET) surface area of the AOPIM-1

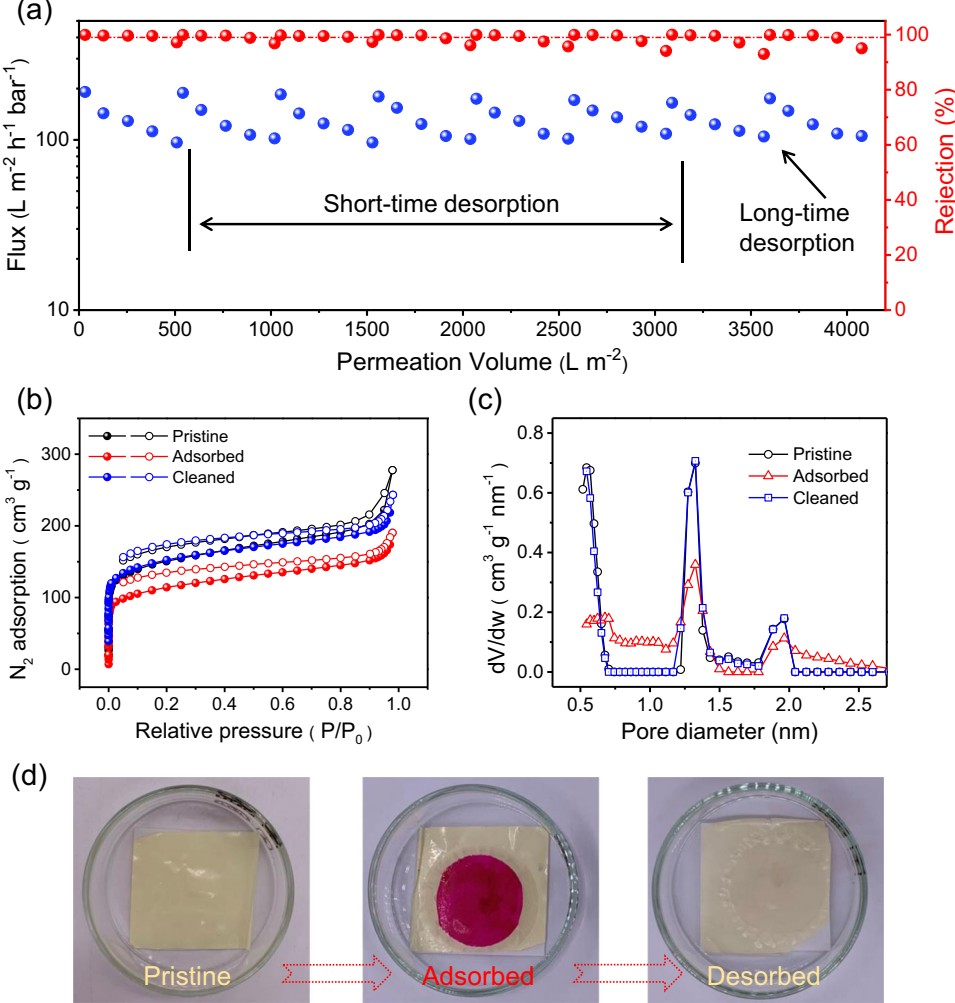

**Fig. 4 Adsorptive separation behavior of AOPIM-1 membrane in a multi-cycle dynamic adsorption-desorption process. a** Flux and RHB removal rate of the membrane in multiple adsorption-desorption cycles (concentration: 20 ppm, applied pressure: 0.2 MPa, pH = 10); **b** Pore size distributions of pristine, adsorbed, and desorbed AOPIM-1 membrane, respectively; **c** Nitrogen absorption-desorption isotherms of pristine, adsorbed, and desorbed AOPIM-1, respectively; **d** optical pictures of the pristine, adsorbed and cleaned AOPIM-1 membrane coupon.

adsorptive membrane is reduced from $552 \, m^2 \, g^{-1}$ to $415 \, m^2 \, g^{-1}$ after the adsorption test, and it can be easily regenerated to the original level after desorption (Fig. 4b). Meanwhile, it can be seen in Fig. 4c, the pore size distribution of AOPIM-1 is also restored to the original level after cleaning. As a control experiment, a negatively charged polyethersulfone ultrafiltration membrane with similar pore size and water flux is selected to filtrate feed solution containing positively charged MB dye molecules. It is found that almost all dye molecules pass through the membrane without noticeable retention (Supplementary Fig. 17), which is common for conventional ultrafiltration membranes that rejects large molecules and particulate matter (such as proteins, suspended solids, bacteria, viruses, and colloids) but cannot accurately separate small organic molecules[12]. Obviously, polymer materials without intrinsic microporosity (polyethersulfone in this case) lack necessary specific surface area and affinity sites to allow the electrostatic attraction between functional groups and target molecules to occur. Based on the above results and discussions, it could be concluded that the electrostatic attraction and the microporous feature of AOPIM-1 membrane co-govern the dynamic adsorption performance of the membrane. The charged molecules electrostatically interact with the charged amidoxime groups inside the AOPIM-1 polymer matrix, and the

microporous structure of AOPIM-1 provides abundant adsorption sites for the high-capacity adsorption of charged molecules during dynamic membrane separations.

**Adsorptive separation of active pharmaceutical ingredients (APIs).** In the pharmaceutical industry, precise separation of organic molecules such as drugs, proteins, and polysaccharides are indispensable for the production of active pharmaceutical ingredients (APIs)[5,7,47]. For instance, the raw material water extracts of phytochemical drugs, an important category of APIs extracted from natural plants[47], usually possess a complex composition including polysaccharides (molecular weights usually range from 10,000 to 100,000 Da), APIs (molecular weight < 2000 Da), and inorganic salts such as sodium chloride. The feasibility of membrane adsorption for the separation of such 3-component systems was evaluated in this section. As schematically illustrated in Fig. 5a and Supplementary Fig. 18, a 2-step process is proposed for adsorptive separation of a phytochemical drug water extract using AOPIM-1 membranes. In the first step, the water extract of natural plants passes through the surface of the membrane, the plant polysaccharides with large molecular weight are trapped on the feed side of the membrane, the active

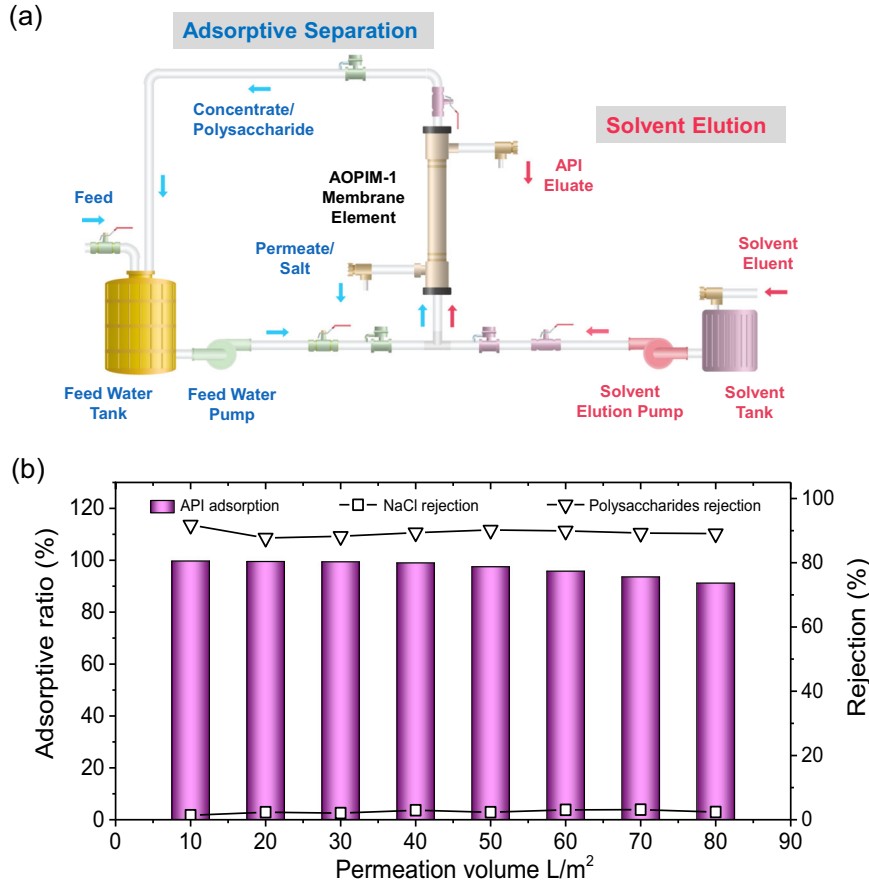

**Fig. 5 Adsorptive separation of active pharmaceutical ingredients (APIs) by AOPIM-1 membranes. a** Schematic illustration of the 2-step process for adsorptive separation of mixed API/polysaccharide/salt feed using AOPIM-1 membranes; **b** The adsorption/rejection ratio of API/polysaccharide/NaCl versus permeation volume of a synthetic water extract feed solution.

pharmaceutical molecules with small molecular weight are adsorbed within the membrane matrix, and inorganic salts are permeated through the membrane. After the membrane is saturated with API adsorption, the second step uses methanol as the eluent to flow in from the feed side, and the API eluate is obtained on the permeate side.

As a proof of concept, an AOPIM-1 membrane with altered casting conditions (3:1 DMF/1,4-dioxane co-solvent[48,49]) and tightened membrane pores (water flux of ~121.3 L m$^{-2}$ h$^{-1}$ bar$^{-1}$, MWCO of ~20 kDa) are prepared, and the MWCO curve of M6 and the pore size distribution are shown in Supplementary Fig. 19. A synthetic water extract (Supplementary Tab. 6) is prepared with Dextran T-200 (MW = 20 kDa) used as the target macromolecular polysaccharide, berberine (MW = 336.4 Da) as the target small API molecule, and sodium chloride as the inorganic salt. As can be seen in Fig. 5b, during a cross-flow filtration cycle, berberine is continuously adsorbed in the membrane, and the rejection rate of Dextran T-200 is maintained above 90%. At the same time, sodium chloride permeates through the membrane without rejection. At the end of one filtration cycle, the feed solution is changed to methanol to elute the berberine enriched in the membrane. Supplementary Fig. 20 clearly shows the UV-vis absorption peaks in the stock solution and filtrate before and after processing 50 mL of the synthetic feed. After the filtration, the characteristic absorption peak of berberine disappeared, which proves that the berberine in the feed solution are completely adsorbed in the membrane. At the same time, it can be seen in the illustration that 50 mL of the feed solution is enriched in the membrane and finally eluted by about

5 mL of methanol, achieving a 10-fold enrichment. Compared with other separation methods, the traditional distillation method is energy-intensive and time-consuming, whereas with the membrane filtration method it is difficult to achieve sufficient accuracy and efficiency, and the obtained permeate usually requires further purification operations afterwards[50]. The membrane adsorption separation method shows the merit of the smallest energy consumption and highest product purity. More importantly, this experiment demonstrates that the ultrahigh processing capacity of AOPIM-1 adsorptive membranes omits the need for frequent cleaning and regeneration, effectively broadening the application feasibility of membrane adsorption from treating trace organic compounds towards mass chemical productions.

This work has developed a type of pH-tunable high-capacity adsorptive membranes based on AOPIM-1, in which the high specific surface area (high adsorption capacity), abundant adsorption sites (adsorption selectivity), reversible adsorbate-adsorbent interaction (fast adsorption/desorption rates), good solubility processability (scale-up feasibility) and hydrophilicity (easy-wet micropores) of AOPIM-1 are fully utilized. The processing capacity and permeability of the membrane is adjusted by manipulating the phase inversion process. While achieving the retention of small molecules, the membrane permeation flux is in line with the level of common ultrafiltration, which is 2 orders of magnitude higher than typical nanofiltration membranes of similar rejections. The best processing capacity reaches 26.114 g m$^{-2}$, which is 10–1000 times higher than the value reported in the literature for existing adsorptive membranes. This

newly developed membrane material realizes a dynamic operation process of membrane adsorption-desorption, cleaning, and regeneration with high efficiency, which can be a good supplement to conventional pressure-driven membrane separation processes. Owing to the superior processing capacity, the AOPIM-1 membrane exhibits high separation efficiency of actual complex systems and guarantees the purity of the product obtained. The development of AOPIM-1 adsorptive membranes thereby broadens the prospects of membrane adsorption for practical applications.

## Methods

**Fabrication of PIM-1 and AOPIM-1 polymers.** Polymer of intrinsic microporosity (PIM-1) was obtained following a previously reported method. Under a nitrogen atmosphere, 3.001 g (15 mmol) tetra-fluoroterephthalonitrile (TFTPN), 5.106 g (15 mmol) 5,5',6,6'-tetrahydroxy-3,3,3',3'- tetramethylspirobisindane (TTSBI) and 30 mL anhydrous DMAc were added into a 100 mL three-necked flask. After the chemicals were completely dissolved, 6.21 g (45 mmol) anhydrous milled K$_2$CO$_3$ was added and the flask was placed into a 160 °C oil bath under mechanical stirring. After approximately 3 min, a viscous yellow solution formed, and 20 mL of toluene was added. Several minutes later, a further 20 mL of toluene was added to dilute the solution. Then, the mixture was poured into 300 mL methanol, and an elastic, threadlike, light-yellow polymer was observed. The polymer product was dissolved in chloroform and reprecipitated in methanol, and then refluxed in Milli-Q water for 4–5 h and dried at 80 °C under vacuum for 48 h.

Hydrophilic amidoxime modified PIM-1 (AOPIM-1) was synthesized by dissolving 0.5 g PIM-1 in 30 mL THF and heating to reflux under N$_2$. Then, 5.0 mL hydroxyl amine was added dropwise, and the solution was further refluxed for 20 h. The resulting polymer was precipitated by the addition of ethanol, filtered, washed thoroughly with ethanol and water, and then dried at 110 °C for 24 h.

**Fabrication of AOPIM-1 membranes.** The polymer dope was obtained by dissolving AOPIM-1 in DMF or DMF/1,4-dioxane mixed solvent, and stirred continuously at room temperature overnight to ensure that the polymer dissolves evenly in the solvent. Then the mixture was left at room temperature for 24 h to remove air bubbles. After that, the polymer solution was used to cast films on a clean glass plate at 25 °C and 40% relative humidity. For the casting solution with co-solvent, the solvent was allowed to evaporate from the surface of the film in 20 s to produce a denser selective skin. Next, the glass plate was immersed into a coagulation bath. After 1 h, membranes were transferred to a fresh water bath and kept for 24 h to finish phase separation. Finally, the membranes were immersed in methanol for future use.

**Membrane material characterization.** The morphology of the as-prepared membranes was observed using a field emission scanning electron microscope (Hitachi S4800, Japan). Before capturing SEM image, a thin Au layer was sputtered onto the membrane under 10 mA for 2 min (Emitech K550X sputtering). $^1$H nuclear magnetic resonance (NMR) spectra were recorded on a Bruker 400 MHz spectrometer using dimethyl sulfoxide-d$_6$ as a solvent. FTIR spectra of membranes were obtained using a Nicolet 6700 FTIR spectrometer (USA). Nitrogen absorption/desorption measurements were performed on a Quantachrome Autosorb IQ-MP-MP at 77 K. All samples were degassed at 120 °C for 12 h before nitrogen absorption measurements were performed. The surface charge of the membrane was determined by streaming potential measurement using a SurPASS 3 electrokinetic analyzer with a flat-plate measuring cell (10 mm × 20 mm). The tensile properties of the membranes were tested by Instron 3365 universal material testing machine and the samples were water-soaked strips with length of 20 mm, width of 10 mm and thickness of 97 μm. The pore size distribution of membrane was measured by Beishide 3H-2000PB filter membrane pore size analyzer.

**Molecular simulation.** All simulations were performed by LAMMPS[51] with the condensed-phase optimized molecular potentials for atomistic simulation studies[52] (COMPASS) force field. The initial charge of molecules was assigned by COMPASS. To find its optimized structure, AOPIM-1 chain and all the small molecular (MB, MO, and H$_2$O) has been annealed from 2000 K to 300 K. AOPIM-1 chains are randomly initiated throughout the simulation box and grown by randomly choosing one of the two chiral monomers and one of the two possible orientations until the target chain length of 12 monomers is achieved or an overlap is the result of adding either monomer type in either orientation. After the initial geometry optimization, the adsorption energy was calculated by: $E_{ads} = E_{AOPIM-1+MO/MB/H_2O} - E_{AOPIM-1} - E_{MB/MO/H_2O}$, where $E_{AOPIM-1+MO/MB/H_2O}$ is the total energy of small molecule (MB/MO/H$_2$O) adsorbed on AOPIM-1, the $E_{AOPIM-1}$ and $E_{MB/MO/H_2O}$ are the energy of AOPIM-1 and small molecules, respectively.

**Static adsorption behavior of AOPIM-1 polymer.** The static adsorption behavior of AOPIM-1 was investigated using dyes with different chargeability (negative,

MO; positive, MB) as model solutes. A dry AOPIM-1 membrane coupon (~10 mg) was placed in dye solution with certain concentration (10–500 mg L$^{-1}$) and pH value. The mixture was stirred continuously at room temperature at least 24 h or a certain time. The concentration of dyes was analyzed by a UV-Vis spectrophotometer (PerkinElmer, Inc.). The amount of dye adsorbed by the AOPIM-1, q (mg g$^{-1}$), was determined from the Eq. (1)[53,54],

$$q = \frac{(C_0 - C_*)V}{w} \tag{1}$$

where $q$ represents the adsorption amount at equilibrium ($q_e$) or the adsorption amount at time t ($q_t$). $C_0$ (mg L$^{-1}$), $C_*$ (mg L$^{-1}$), $V$ (L), and $W$ (g) represent initial concentration of dye, concentration of dye at equilibrium or time t, volume of solution and amount of AOPIM-1, respectively.

The effect of pH on the adsorption was studied by adjusting pH of the dye solutions to 3–10 with the help of 0.1 M NaOH and 0.1 M HCl.

The adsorption kinetics of dyes by AOPIM-1 was studied using a second-order equation in nonlinear form by Eq. (2)[55,56],

$$q_t = \frac{k_2 q_e^2 t}{1 + k_2 q_e t} \tag{2}$$

where $k_2$ (g mg$^{-1}$ min$^{-1}$) is the second-order rate constant. The adsorption capacity of dyes by AOPIM-1 was studied using a Langmuir isotherm model by Eq. (3)[55,56],

$$q_e = \frac{K_L q_m C_e}{1 + K_L C_e} \tag{3}$$

where $K_L$ (L mg$^{-1}$) represents the affinity constant and $q_m$ (mg g$^{-1}$) is the maximum adsorption capacity of the adsorbate.

**Dynamic adsorptive separation properties of AOPIM-1 membranes.** The dynamic adsorption properties of AOPIM-1 membranes were investigated using six types of dyes with different chargeability (negative: MO, CR, BB; positive: MB, RHB, CV). Filtration experiments were performed in a dead-end filtration cell with an effective membrane area of 3.14 cm$^2$ (Supplementary Fig. 21). Each tested membrane was compacted by the filtration of deionized water under 0.2 MPa for 2 h in order to achieve a steady flux. Then, the permeate flux (J (L m$^{-2}$ h$^{-1}$ bar$^{-1}$)) used different dyes as feed solution (20 mg L$^{-1}$, PH = 3/10) was measured under 0.2 MPa at room temperature and calculated using the following Eq. (4),

$$J = \frac{V}{\Delta PAt} \tag{4}$$

where $\Delta P$ (bar), $V$ (L), $A$ (m$^2$), and $t$ (h) represent permeate flux, transmembrane pressure, permeate volume, membrane area and filtration time, respectively. The rejection rate (R (%)) of dyes was calculated from Eq. (5),

$$R = 1 - \frac{C_P}{C_F} \tag{5}$$

where $C_P$ and $C_F$ correspond to the dye concentrations in the permeate and feed solutions, respectively. The dye concentrations in the permeate and feed solutions were determined using a UV/Vis spectrometer (Biochrom Libra S32). The processing capacity used RHB as feed solution (20 mg L$^{-1}$, PH = 10) was measured under 0.2 MPa at room temperature and determined by permeation volume when rejection is higher than 99%.

The effects of dye concentration on membrane separation performance were performed in a dead-end filtration cell with an effective membrane area of 3.14 cm$^2$, and used RHB as feed solution (10/20/30/50 mg L$^{-1}$, pH = 10) were measured under 0.2 MPa at room temperature.

The multi-cycle adsorptive separation performance stability of the AOPIM-1 adsorptive membrane was performed in a dead-end filtration cell with an effective membrane area of 3.14 cm$^2$ for 8 cycles. For a typical cycle, the RHB solution (20 mg L$^{-1}$, pH = 10) was filtrated through the membrane for 500 L m$^{-2}$ at 0.2 MPa, and the permeate flux (J) was calculated by Eq. (4). Afterwards, the membrane is subjected to a short time (five minutes) desorption process by switching the feed solution of the membrane to about 20 ml methanol for cleaning, and the eluent can be obtained at the outlet. After that, fresh feed dye solution was applied for the next cycle.

**Adsorptive separation of APIs by AOPIM-1 membranes.** Separation of a synthetic 3-conponent mixture feed representing inorganic salt, polysaccharides and active pharmaceutical ingredient (API) chemicals in the water extract of natural plants was performed in a lab-scale cross-flow cell (Supplementary Fig. 20) at room temperature. The feed solution included NaCl, dextran T-200 and berberine with detailed composition listed in Supplementary Tab. 6. The concentrations of each substance in the feed and permeate solution were measured by conductometer (FE30K, Mettler Toledo), total organic carbon (TOC) analyzer (Aurora 1030 W) and UV-vis spectrophotometer (PerkinElmer, Inc.), respectively.

## Data availability

The authors declare that the data supporting the findings of this study are available within the paper [and its Supplementary Information files]. Any additional detail can be requested from the corresponding authors.

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

## Acknowledgements

This work was supported by the National Key R&D Program of China (2019YFA0705800), the National Natural Science Foundation of China (21988102, 51803145, 51873230), Funding support from the Social Development Program of Jiangsu Province (BE2019678) is gratefully appreciated as well.

## Author contributions

Z.W., W.F. and J.J. conceived the initial idea and experimental design. Z.W., X.L. and Z.S. performed the membrane fabrication and the main characterization experiments. S.Z. and Y.Y. performed the gas sorption experiment of AOPIM-1 membrane. K.L. carried out the molecular dynamics simulations and analyzed the data. Z.W., X.L., Y.Z., W.F. and J.J. wrote the manuscript. All authors analyzed results and commented on the revised manuscript.

## Competing interests

The authors declare no competing interests.
