## [Peer Review File · Nature Communications]

Microporous Polymer Adsorptive Membranes with High Processing Capacity for Molecular SeparationReviewers' Comments:

Reviewer #1:

Remarks to the Author:

The amidoxime-modified polymer of intrinsic microporosity AOPIM-1 has recently attracted much attention for a variety of applications. This manuscript highlights its potential for use in adsorptive membranes.

There is a key reference missing from the manuscript: B. Satilmis in Journal of Polymers and the Environment (2020) 28:995–1009 (<https://doi.org/10.1007/s10924-020-01664-4>) has previously reported adsorption of methylene blue and methyl orange by AOPIM-1. This should be discussed in relation to the present work.

Previous work on adsorptive membranes with these types of polymers has generally focused on membranes formed by electrospinning, which may be a difficult technology to scale up. In contrast, this work used readily-scalable conventional phase inversion techniques to create highly effective membranes.

This work demonstrates the effectiveness of the membrane at adsorbing different types of dye at different pH values, over a number of adsorption/desorption cycles. It further demonstrates an ingenious process for separating active pharmaceutical ingredients from both high molecular weight polysaccharides and inorganic salts.

Thus, while neither the polymer nor the method of membrane fabrication are entirely novel, this work represents an important advance in their mode of application.

There are a number of minor corrections required:

Line 62: "merit" should be "merits"

Line 92: "free volumes" should perhaps be "free volume elements"

Line 102: "comparable" should be "is comparable"

Line 117: "552.3 m² g⁻¹" It is not justifiable to quote this with a precision of four significant figures

Lines 124,125: "stability ... are" should be "stability ... is"

Line 174: "resulted" should be "the resulting"

Line 175: "processibility capacity" should probably be "processing capacity"

Line 181: "The increasing in" should be "The increase in"

Line 198: "Methyl Blue (MB)" should probably be "Methylene Blue (MB)" as in line 139. Methylene Blue (C₁₆H₁₈CIN₃S) and Methyl Blue (C₃₇H₂₇N₃Na₂O₉S₃) are different dyes.

Line 244: "reduced" should be "reduction in"

Lines 289,290: "whereas the membrane filtration method is difficult" should be "whereas with the membrane filtration method it is difficult"

Line 296: "broadened" should be "broadening"

Line 316: "highly guarantees" should be just "guarantees"

Reviewer #2:

Remarks to the Author:

This manuscript reports a hydrophilic amidoxime modified polymer of intrinsic micro porosity (AOPIM-1) as a membrane adsorption material to selectively adsorb and separate small organic molecules from water with ultrahigh processing capacity. This work showed an impressed membrane for efficient molecule separation, especially for the real compound extraction from water. However, there are still some problems in this work that need to be solved.

1. As we know that the major advantage of membrane is filtration, while compare with traditional adsorbents like powders and particles the adsorption capacity of membrane is not at the front of the line. The author should expound the necessity and advantage for membrane adsorbents.
2. From the design of the PIM, apart from the electrostatic interactions, H-bonding and conjugated effect contribute to the dye molecules sieving. The author should provide more explanations.
3. For the mechanism part, the molecular dynamics simulations on polymers with water or dye molecules separation might be supported as you have highlighted in your author contribution part.
4. The core size distribution of the prepared membrane should be given as it affects the permeance of the membrane.
5. How does the concentration of the solutes influence the separation performance of the membrane?
6. How to understand the efficient membrane separation? Some related work should be useful to understand this. (e.g. Nature communications 2018, 9 (1), 1-8; Journal of Materials Chemistry A 2018, 6 (42), 21104-21109).
7. Although the adsorption capacities for ionic dyes were outstanding in acidic or alkaline conditions, the adsorption capacity in neutrally and practical condition is more meaningful, because it is impossible or too difficult to adjust the pH values of the wastewater during the application. The major adsorption studies and analysis should be performed in neutrally condition. Meanwhile, for the comparison with other works, the adsorption capacities in this work should be the values in neutrally condition.
8. For the removal of ionic dyes with membranes, the following works should also be compared: J. Colloid Interf. Sci., 2019, 556, 492-502; J. Mater. Chem. A, 2018, 6, 13359-13372; J. Mater. Sci. Technol., 2021, 78, 131-143.
9. The mechanical properties like tensile strength of the membrane should be tested.

Reviewer #3:

Remarks to the Author:

The manuscript reports the development of microporous polymer as adsorptive membranes for separation applications. PIM polymers are an important class of polymers for separation applications. Conventional PIMs are difficult to process due to their solubility in a limited range of solvents. Recent studies on amidoxime-functionalized PIMs have shown great promise owing to their solubility in polar solvents such as DMF, DMSO, NMP. Therefore, solution processing of these polymers by non-solvent-induced phase separation (NIPS) into hierarchically porous membranes is quite novel. This manuscript reports the processing of AO-PIM polymer into sponge-like porous membranes and demonstrated their application as membrane adsorbents. Compared to previous work on membrane adsorbents such as MOF-based membranes, these PIM membranes show a great advantage in terms of processability and functionality. The authors demonstrated the selective removal of dye molecules in acidic and alkaline solutions. These adsorptive membranes provide adsorption functions that cannot be achieved by conventional nanofiltration and ultrafiltration membranes. The demonstration of separation of API shows promising potential of these membranes. While these membranes show impressive apparent adsorption capacity and high water flux, the data interpretation and comparison need to be considered carefully because the membranes work as membrane adsorbents instead of separation membranes.

The manuscript is suitable for publication in Nature Communications, but some technical concerns should be addressed.

There are some concerns about the technical aspects of the results and analyses.

1. It is an innovative idea to tune the charge of the polymer by protonation and deprotonation of the AO groups and use the membranes to separate anionic and cationic dyes. One technical problem is that the protonation of AO groups in acidic solution could lead to significant decay of mechanical properties. Did the authors measure the mechanical properties of membranes treated at low pH?

2. The adsorption mechanisms need in-depth study and analysis. It is known that for these microporous materials with charges, the adsorption of dye molecules may be dominated by electrostatic adsorption on the surface of membranes instead of purely adsorption in the micropores. The schematic diagram shown in Fig. 1c is likely incorrect. Do you have direct evidence to claim that all the dye molecules are adsorbed in the micropores?

3. The adsorption capacity presented in Figure 3e needs to be explained carefully. Because the molecules are likely adsorbed on the surface of membranes and form a cake layer, the adsorption of these molecules are apparent values instead of intrinsic values. Also the capacity should be linked with the pore volume of the membranes available for adsorption instead of membrane area, because the thickness of these membranes are different. So the adsorptive capacity unit should be changed to g/m^3 instead of g/m^2 .

The authors tried to compare the rejection and flux trade-off with conventional separation membranes based on size and charge. In my opinion, this is not a fair comparison. When the porous polymers are used as adsorbents, the separation performance are dynamically changing, both rejection and water flux are dynamic values. Therefore, the adsorption performance in terms of flux and rejection are not intrinsic properties of the membranes. If the authors intend to compare the performance with other membranes, they should clarify that these membranes are adsorbents, and compare with previously reported membrane adsorbents. For example, MOF membrane adsorbents show water permeance of $20,000 \text{ L m}^{-2} \text{ bar}^{-1} \text{ h}^{-1}$, which is much higher than the water permeance achieved in this study. The MOF membrane adsorbent also showed high rejection at the beginning of filtration tests.

References: High-permeance metal-organic framework-based membrane adsorbent for the removal of dye molecules in aqueous phase. *Environ. Sci.: Nano*, 2017,4, 2205-2214.

4. The multicycle dynamic adsorption and desorption process shown in Figure 4 needs to be carefully explained. Normally, for these adsorbents, as the high-surface area membranes adsorb more molecules and reach equilibrium, the membrane will lose adsorption capacity and the membrane fouling will become dominant. The authors explained that the molecules adsorbed on the surface and inside of the membrane, and likely the water flux decreases. In this case, the phenomenon is membrane fouling. To avoid confusion, the authors should report the original data of the filtration, particularly, the dye concentration of the feed, retentate, and the permeate, so you can estimate the mass balance.

5. It is recommended to compare the adsorption separation performance with control experiments using AO-PIM powders. For example, you can prepare AO-PIM particles with hierarchical porosity and load the particles into a column and study the adsorption and separation performance.

6. The authors need to change some of the expressions, such as extremely high adsorption capacity, unprecedented capacity.

7. The authors are recommended to perform a deeper literature search, and include relevant reports in the references, for example, recent report on preparation of hierarchical AO-PIM membranes by NIPS method. *Nature Sustainability*, 5, 71-80 (2022).

8. To some extent, the AO-PIM membrane reported by the authors are still ultrafiltration membranes. It will be good to compare with recent development of ultrafiltration membranes in the literature. Some recent studies on ultrafiltration membranes also show very high water flux and high rejection towards dye molecules. For example, recent work published in *Separation and Purification Technology*, 2022, 283, 120163, showed, a pure water flux of 110.4 LMH with 99.2% rejection to dye molecules (Congo red).

9. Figure S11, it would be better to provide more analysis of the dye molecules, for example, molecular volume and size.

Reviewer #1: The amidoxime-modified polymer of intrinsic microporosity AOPIM-1 has recently attracted much attention for a variety of applications. This manuscript highlights its potential for use in adsorptive membranes.

There is a key reference missing from the manuscript: B. Satilmis in *Journal of Polymers and the Environment* (2020) 28:995–1009 (<https://doi.org/10.1007/s10924-020-01664-4>) has previously reported adsorption of methylene blue and methyl orange by AOPIM-1. This should be discussed in relation to the present work.

Previous work on adsorptive membranes with these types of polymers has generally focused on membranes formed by electrospinning, which may be a difficult technology to scale up. In contrast, this work used readily-scalable conventional phase inversion techniques to create highly effective membranes.

This work demonstrates the effectiveness of the membrane at adsorbing different types of dye at different pH values, over a number of adsorption/desorption cycles. It further demonstrates an ingenious process for separating active pharmaceutical ingredients from both high molecular weight polysaccharides and inorganic salts.

Thus, while neither the polymer nor the method of membrane fabrication are entirely novel, this work represents an important advance in their mode of application.

There are a number of minor corrections required:

Line 62: “merit” should be “merits”

Line 92: “free volumes” should perhaps be “free volume elements”

Line 102: “comparable” should be “is comparable”

Line 117: “552.3 m² g⁻¹” It is not justifiable to quote this with a precision of four significant figures

Lines 124,125: “stability ... are” should be “stability ... is”

Line 174: “resulted” should be “the resulting”

Line 175: “processibility capacity” should probably be “processing capacity”

Line 181: “The increasing in” should be “The increase in”

Line 198: “Methyl Blue (MB)” should probably be “Methylene Blue (MB)” as in line 139. Methylene Blue (C₁₆H₁₈ClN₃S) and Methyl Blue (C₃₇H₂₇N₃Na₂O₉S₃) are different dyes.

Line 244: “reduced” should be “reduction in”

Lines 289,290: “whereas the membrane filtration method is difficult” should be “whereas with the membrane filtration method it is difficult”

Line 296: “broadened” should be “broadening”

Line 316: “highly guarantees” should be just “guarantees”.

Response: Thanks a lot for the reviewer’s positive comment and kind suggestion on revising our manuscript.

According to the reviewer’s suggestion, the work by Bekir et al. (<https://doi.org/10.1007/s10924-020-01664-4>) has been added in the reference as ref. 40, and the following discussion has been included in the revised manuscript (page 8, line 4-8): “Bekir et al. also reported the adsorption of charged dyes, methylene blue (MB) and methyl orange (MO), from water system by AO-PIM-1 powder. The experimental adsorption capacities of AO-PIM-1 are 79.8 mg/g and 69.8 mg/g for MO and MB at pH 6, respectively, which is consistent with our data.”

As for the minor corrections pointed by the reviewer, we have all corrected and highlighted them in the manuscript.

Reference:

Ref 40. Satilmis, B. Amidoxime Modified Polymers of Intrinsic Microporosity (PIM-1); A Versatile Adsorbent for Efficient Removal of Charged Dyes; Equilibrium, Kinetic and Thermodynamic Studies. *J. Polym. Environ.* **28**, 995-1009 (2020).

Reviewer #2 : This manuscript reports a hydrophilic amidoxime modified polymer of intrinsic micro porosity (AOPIM-1) as a membrane adsorption material to selectively adsorb and separate small organic molecules from water with ultrahigh processing capacity. This work showed an impressed membrane for efficient molecule separation, especially for the real compound extraction from water. However, there are still some problems in this work that need to be solved.

Reviewer's comment: 1) As we know that the major advantage of membrane is filtration, while compare with traditional adsorbents like powders and particles the adsorption capacity of membrane is not at the front of the line. The author should expound the necessity and advantage for membrane adsorbents.

Response: Thanks for the kind suggestion from the reviewer. Indeed, traditional adsorption techniques with large amount of powder or particle adsorbents exhibits high adsorption capacity and separation selectivity. But it also has some major drawbacks like low processing rate, high internal diffusion resistance within adsorbents etc. Membrane adsorbents could achieve fast processing of a large volume of feed solution in a dynamic separation process, which is a key advantage over power adsorbents. Moreover, as the application of membrane adsorption is hindered by the lack of adsorptive membranes with sufficient processing capacity, this work specifically provides a membrane adsorbent with high capacity that fulfils the necessity of practical membrane adsorption.

According to the reviewer's suggestion, the necessity and advantage for membrane adsorbents has been emphasized in the revised manuscript (page 3, line 16-22; page 4, line5-11): “In contrast, adsorption can be a highly selective molecular separation process with specific physical or chemical interactions between adsorbents and target molecules¹³⁻¹⁶, but its application is often limited by its low processing rate, high internal diffusion resistance within adsorbents, etc¹⁷⁻¹⁸. Membrane adsorption is a pressure-driven dynamic membrane-based adsorption process, which combines the merits of both adsorption and membrane separation¹⁹⁻²¹.... Therefore, they are expected to break through the permeability-selectivity trade-off by simultaneously achieving the selectivity of dense membranes and permeability of porous membranes^{20,25-28}. However, the application of membrane adsorption is hindered by the lack of adsorptive membranes with sufficient processing capacity, which greatly affects their further development and practical application.”.

Reviewer's comment: 2) From the design of the PIM, apart from the electrostatic interactions, H-bonding and conjugated effect contribute to the dye molecules sieving. The author should provide more explanations.

Response: Thanks for reviewer's keen questions. As shown in Figure R1, AOPIM-1 favors the capture of negatively charged MO molecules under acidic conditions while capturing the oppositely charged MB under alkaline conditions, exhibiting a pH-tunable adsorption feature. And much lower equilibrium adsorption capacity is observed under neutral pH conditions where the amidoxime possess minimal chargeability, which reveals that the pH-tunable affinity sites make major contribution to the adsorption capacity. Therefore, from the interpretation of experimental data, the dye molecules sieving by the AOPIM-1 adsorptive membrane should be mainly attributed to electrostatic interactions.

Figure R1. Equilibrium adsorption capacity (q_e) of AOPIM-1 for MO and MB dye molecules at different pH conditions.

The discussion regarding Figure 2b has been revised as follows (page 7, line 22; page 8, line 1-4): “...And much lower equilibrium adsorption capacity is observed under neutral pH conditions where the amidoxime possess minimal chargeability, which reveals that the pH-tunable affinity sites make major contribution to the adsorption capacity, and that the dye molecules sieving should be mainly attributed to electrostatic interactions.”.

Reviewer's comment: 3) For the mechanism part, the molecular dynamics simulations on polymers with water or dye molecules separation might be supported as you have highlighted in your author contribution part.

Response: According to the reviewer's suggestion, the molecular dynamics simulations has been conducted on polymers with water and dye molecules (MB and MO) separation in alkaline condition. Energy optimized molecular model and adsorption energies of AOPIM-1 to MB, MO and H₂O molecule in alkaline condition are shown in Figure R2. The absorption energy (E_{ads}) of AOPIM-1 to MB, MO and H₂O molecule are -1587.7, -745.4, and -742.3 kcal/mol, respectively. Obviously, the absorption energy (E_{ads}) of AOPIM-1 to MB is the highest, which is very consistent with the experimental result that MB molecule was selectively adsorbed and separated by the AOPIM-1 membrane.

The simulation method and results summarized as follows and the discussion “Energy optimized molecular model and adsorption energies of AOPIM-1 to MB, MO and H₂O molecule in alkaline condition are shown in Figure S8. The absorption energy (E_{ads}) of AOPIM-1 to MB, MO and H₂O molecule are -1587.7, -745.4, and -742.3 kcal/mol, respectively. Obviously, the absorption energy (E_{ads}) of AOPIM-1 to MB is the highest, which is very consistent with the experimental result that MB molecule was selectively adsorbed and separated by the AOPIM-1 membrane.” have been added in the revised manuscript (page 8, line 14-20) and the revised supporting information as Figure S8.

Simulation method

All simulations were performed by LAMMPS¹ with the condensed-phase optimized molecular potentials for atomistic simulation studies² (COMPASS) force field. The initial charge of molecules was assigned by COMPASS. To find its optimized structure, AOPIM-1 chain and all the small molecular (MB, MO, and H₂O) has been annealed from 2000 K to 300 K. AOPIM-1 chains are randomly initiated throughout the simulation box and grown by randomly choosing one of the two chiral monomers and one of the two possible orientations until the target chain length of 12 monomers is achieved or an overlap is the result of adding either monomer type in either orientation. After the initial geometry optimization, the adsorption energy was calculated by: $E_{ads} = E_{AOPIM-1+MO/MB/H_2O} - E_{AOPIM-1} - E_{MB/MO/H_2O}$, where $E_{AOPIM-1+MO/MB/H_2O}$ is the total energy of small molecule (MB/MO/H₂O) adsorbed on AOPIM-1, the $E_{AOPIM-1}$ and $E_{MB/MO/H_2O}$ are the energy of AOPIM-1 and small molecules, respectively.

Reference

61. Plimpton, S. Fast parallel algorithms for short-range molecular dynamics. *J. Comput. Phys.* **117**, 1-19 (1995).
62. Sun, H. Compass: an ab initio force-field optimized for condensed-phase applications overview with details on alkane and benzene compounds. *J. Phys. Chem. B.* **102**, 7338-7364 (1998).

Figure R2. Energy optimized molecular model and adsorption energies of AOPIM-1 to MB, MO and H₂O molecule.

Reviewer's comment: 4) The core size distribution of the prepared membrane should be given as it affects the permeance of the membrane.

Response: According to the reviewer's suggestion, we measured the pore size distribution of membranes prepared in different coagulation baths by the bubble point method. As shown in Figure R3, it can be seen that the mean pore size of M1, M2 and M3 is 271 nm, 247 nm and 211 nm, respectively. The pore size decreases with increasing ethanol content in the coagulation bath. This is also consistent with the variation trend of membrane flux. The pore size distribution of the membranes has been added in the supporting information as Figure S9. To achieve API separation, a tightened membrane pores (water flux of $\sim 121.3 \text{ L m}^{-2} \text{ h}^{-1} \text{ bar}^{-1}$, MWCO of $\sim 20 \text{ kDa}$) are prepared, and the mean pore size of M6 is 3 nm as shown in Figure R4 (Figure S18 in supporting information).

Figure R3. Pore size of membranes prepared in different coagulation baths.

Figure R4. MWCO curve and pore size distribution of M6.

Reviewer's comment: 5) How does the concentration of the solutes influence the separation performance of the membrane?

Response: To address the reviewer's query regarding the effect of solution concentration on the membrane, the separation performance of M3 membrane were

measured by using 10/20/30/50 mg L⁻¹ RHB feed solution. As the results shown in the figure below, the permeation volume (rejection > 99%) of 10/20/30/50 mg L⁻¹ RHB solution is 1067.6, 628, 251.2, 125.6 L m⁻², respectively. And the flux of the 10/20/30/50 mg L⁻¹ RHB solution varies from 297.79 L m⁻² h⁻¹ bar⁻¹ to 292.12 L m⁻² h⁻¹ bar⁻¹, from 288.79 L m⁻² h⁻¹ bar⁻¹ to 283.57 L m⁻² h⁻¹ bar⁻¹, from 276.93 L m⁻² h⁻¹ bar⁻¹ to 270.24 L m⁻² h⁻¹ bar⁻¹ and from 272.97 L m⁻² h⁻¹ bar⁻¹ to 269.4 L m⁻² h⁻¹ bar⁻¹. The mean flux slightly decreases with the increase of RHB solution concentration, while the total adsorption capacity of the adsorptive membrane is almost unchanged (11.2-12.7 g m⁻²). It is obvious that the concentration of solutes have very little influence on the separation performance of our adsorptive membrane.

The relevant data and discussion have been added in the revised supporting information (Figure S13): “The solutes concentration influence on the membrane separation performance is further investigated by separating 10/20/30/50 mg L⁻¹ RHB solution. In the Figure S13, it can be seen that the flux of the 10/20/30/50 mg L⁻¹ RHB solution varies from 197.21 L m⁻² h⁻¹ bar⁻¹ to 193.46 L m⁻² h⁻¹ bar⁻¹, from 191.25 L m⁻² h⁻¹ bar⁻¹ to 187.79 L m⁻² h⁻¹ bar⁻¹, from 183.39 L m⁻² h⁻¹ bar⁻¹ to 178.97 L m⁻² h⁻¹ bar⁻¹ and from 180.78 L m⁻² h⁻¹ bar⁻¹ to 178.44 L m⁻² h⁻¹ bar⁻¹. And the permeation volume (rejection > 99%) of 10/20/30/50 mg L⁻¹ RHB solution is 1067.6, 628.1, 251.2, 125.6 L m⁻², respectively. The mean flux of membrane slightly decreases with increase of RHB solution concentration, while the adsorption capacity is almost unchanged (11.2-12.7 g m⁻²). Obviously, the concentration of solutes has very little influence on the separation performance of our adsorptive membrane.”

Figure R5. Effects of different concentrations of dyes on (a) flux, (b) rejection, and (c) processing capacity.

Reviewer's comment: 6) How to understand the efficient membrane separation? Some related work should be useful to understand this. (e.g. Nature communications 2018, 9 (1), 1-8; Journal of Materials Chemistry A 2018, 6 (42), 21104-21109).

Response: According to the reviewer's reference and our understanding, an efficient separation membrane for industrial filtration and separation processes should be endowed with high flux and high selectivity capability and be stable in aqueous and various organic solvents, but should also be stable in harsh environments. Our AOPIM-1 membrane achieves >99.9% removal of various nano-sized organic molecules with water flux 2 orders of magnitude higher than typical pressure-driven membranes of similar rejections. And they are very stable in various pH condition and behave ultrahigh adsorptive capacity, which could be regarded as an efficient separation membrane. The references provided by the reviewer are very helpful for the understanding of the topic, and they have been referred (9-10) in the revised manuscript (page 3, line 8-10): "However, trade-off between membrane permeability and selectivity is an inherent shortcoming of membrane-based molecular separation"⁷⁻¹⁰ .

Reference

Ref 9. Golalikhani, M., Lei, Q., Chandrasena, R. U., Kasaei, L., Park, H., Bai, J., Orgiani, P., Ciston, J., Sterbinsky, G. E., Arena, D. A., Shafer, P., Arenholz, E., Davidson, B. A., Millis, A. J., Gray, A. X., & Xi, X. X. Nature of the metal-insulator transition in few-unit-cell-thick LaNiO₃ films. *Nat. Commun.* **9**, 1-8 (2018).

Ref 10. Chen, C., Liu, D., Wang, J., Wang, L., Sun, J., & Lei, W. Functionalized boron nitride membranes with multipurpose and super-stable semi-permeability in solvents. *J Mater. Chem. A.* **6**, 21104-21109 (2018).

Reviewer's comment: 7) Although the adsorption capacities for ionic dyes were outstanding in acidic or alkaline conditions, the adsorption capacity in neutrally and practical condition is more meaningful, because it is impossible or too difficult to adjust the pH values of the wastewater during the application. The major adsorption studies and analysis should be performed in neutrally condition. Meanwhile, for the comparison with other works, the adsorption capacities in this work should be the values in neutrally condition.

Response: We could not agree with the reviewer that the adsorption capacity in neutral condition is more meaningful. Wastewater could have very different characters in different industries, it is common to encounter acidic or alkaline wastewater. Taking textile wastewaters as an example, their pH values are easily under 2 or above 12, as the textile is frequently treated in acidic or alkaline conditions to enhance color fixing capability of dyes. Besides wastewater treatment, other industrial separation processes like active pharmaceutical ingredient extraction, food industry concentration and purification, etc., a lot of target feed solutions possess acidic or alkaline pH conditions. Our amidoxime modified polymer favors the capture of negatively charged molecules under acidic conditions while capturing the oppositely charged molecules under alkaline conditions. It demonstrates a pH-tunable adsorption feature, which could be applied in various pH conditions for different dyes

treatment. This unique character of AOPIM-1 membrane is also the highlight of this work.

Regarding the separation behavior of AOPIM-1 membranes in neutral pH environment, the static adsorption capacities at various pH values has been demonstrated in our manuscript (Figure 2b). The adsorption capacity of the polymer is found to be as high as 445.02 mg g^{-1} (MO, pH = 3.3) and 735.75 mg g^{-1} (MB, pH = 10.9), respectively. And lower equilibrium adsorption capacity ($70\text{-}120 \text{ mg g}^{-1}$ for MB and MO) is observed under neutral pH conditions. Furthermore, the dynamic adsorption capacity of our membrane in neutral pH condition is measured. As shown in the Figure below, the membrane adsorption capacity in pH 7 is 0.3 and 7.1 g/m^2 for MO and RHB, respectively, much lower than that in acid or alkaline condition. This is ascribed to the minimal chargeability in neutral condition. It also illustrates that the pH-tunable affinity sites make dominant contribution to the adsorption capacity of our membrane. The relative discussion and result have been added in the revised manuscript (page 11, line 16-21) and supporting information (Figure S15).

Figure R6. Equilibrium adsorption capacity (q_e) of AOPIM-1 for MO and MB dye molecules at different pH conditions;

Figure R7. Membrane processing capacity of RHB in neutral and alkaline condition.

Reviewer's comment: 8) For the removal of ionic dyes with membranes, the following works should also be compared: *J. Colloid Interf. Sci.*, 2019, 556, 492-502; *J. Mater. Chem. A*, 2018, 6, 13359-13372; *J. Mater. Sci. Technol.*, 2021, 78, 131-143.

Response: Thanks for reviewer's kind advice. The works mentioned by the reviewer has been included in Table S1 for comparison.

Table S1. Comparison of adsorption capacity of various adsorbents reported in the literature.

Adsorbents	Chargeability	Specific surface area (m ² g ⁻¹)	Adsorption capacity (mg g ⁻¹)	Adsorbate	Ref.
NH ₂ -UIO-66	Positive	1035	697.7	CR	20
MIL-100 (Cr)	Negative	3100	507.7	CV	20
ZIF-8@CS sponge	Negative	-	987.0	CR	42
Hydrolyzed PIM-1	Positive	-	424.8	MB	34
			42.3	MO	
PIM-1	Neutral	786	~4	MB	34
			~1	MO	
Ethanolamine modified PIM-1	Negative	--	525	Acid Red I	43
AOPAN nanofibrous	pH-response	--	~72	MO	44
			86.7	MO	
AOPIM-1	Neutral	--	81.3	MB	40
PES nanofibrous membrane	Positive	20	208	CR	22
PES/MS	Neutral	--	602.3	MB	46
PQAM nanofibrous membrane	Positive	--	909.8	MO	45

AOPIM-1	pH-response	550	491.63	MO	This work
			765.09	MB	

Reference

22. Lv, C., Chen, S., Xie, Y., Wei, Z., Chen, L., Bao, J., He, C., Zhao, W., Sun, S., & Zhao, C. Positively-charged polyethersulfone nanofibrous membranes for bacteria and anionic dyes removal. *J. Colloid Interface Sci.* **556**, 492-502 (2019).

45. Bao, J., Li, H., Xu, Y., Chen, S., Wang, Z., Jiang, C., Li, H., Wei, Z., Sun, S., Zhao, W., & Zhao, C. Multi-functional polyethersulfone nanofibrous membranes with ultra-high adsorption capacity and ultra-fast removal rates for dyes and bacteria. *J. Mater. Sci. Technol.* **78**, 131-143 (2021).

Ref 46. Xu, Y., Yuan, D., Bao, J., Xie, Y., He, M., Shi, Z., Chen, S., He, C., Zhao, W., & Zhao, C. Nanofibrous membranes with surface migration of functional groups for ultrafast wastewater remediation. *J. Mater. Chem. A.* **6**, 13359-13372 (2018).

Reviewer's comment: 9) The mechanical properties like tensile strength of the membrane should be tested.

Response: According to the reviewer's suggestion, we have tested the tensile strength of AOPIM-1 adsorption membrane under different pH values and calculated their tensile modulus respectively. The tensile stress at break is 2.7, 3.1 and 3.3 MPa, and the modulus is 20.5, 47.9, 54.0 MPa in pH=3, 7 and 10, respectively. The mechanical strength at pH=3 is the lowest, likely due to the decreased H-bond interaction in acid pH environment.

The relevant discussion has been added in the revised manuscript (page 6, line 21-22; page 7, line 1-2): "All the membranes in various pH condition demonstrate sufficient mechanical strength for pressurized permeation tests (Figure S6), while the mechanical strength appears lower in acid condition, likely due to decreased interchain H-bond interaction."

Figure R8. Mechanical properties of AOPIM-1 adsorption membranes under different treatment conditions.

Reviewer #3: The manuscript reports the development of microporous polymer as adsorptive membranes for separation applications. PIM polymers are an important

class of polymers for separation applications. Conventional PIMs are difficult to process due to their solubility in a limited range of solvents. Recent studies on amidoxime-functionalized PIMs have shown great promise owing to their solubility in polar solvents such as DMF, DMSO, NMP. Therefore, solution processing of these polymers by non-solvent-induced phase separation (NIPS) into hierarchically porous membranes is quite novel. This manuscript reports the processing of AO-PIM polymer into sponge-like porous membranes and demonstrated their application as membrane adsorbers. Compared to previous work on membrane adsorbers such as MOF-based membranes, these PIM membranes show a great advantage in terms of processability and functionality. The authors demonstrated the selective removal of dye molecules in acidic and alkaline solutions.

These adsorptive membranes provide adsorption functions that cannot be achieved by conventional nanofiltration and ultrafiltration membranes. The demonstration of separation of API shows promising potential of these membranes. While these membranes show impressive apparent adsorption capacity and high water flux, the data interpretation and comparison need to be considered carefully because the membranes work as membrane adsorbers instead of separation membranes. The manuscript is suitable for publication in Nature Communications, but some technical concerns should be addressed.

There are some concerns about the technical aspects of the results and analyses.

Reviewer's comment: 1) It is an innovative idea to tune the charge of the polymer by protonation and deprotonation of the AO groups and use the membranes to separate anionic and cationic dyes. One technical problem is that the protonation of AO groups in acidic solution could lead to significant decay of mechanical properties. Did the authors measure the mechanical properties of membranes treated at low pH?

Response: According to the reviewer's suggestion, we have measured the mechanical properties of membrane treated at low pH, and compared with those of membrane treated at higher pH as shown in Figure R8. The tensile stress at break of membrane treated at low pH is 2.7 Mpa. The mechanical strength of membrane treated at pH=3 is lower than those of membrane treated at higher pH (3.1 Mpa for pH=7 and 3.3 Mpa for pH=10). This is mainly attributed to protonation of AO groups in acidic solution, resulting the decreased interchain H-bond interaction. In spite of this, the membranes treated at low pH still have strong-enough mechanical strength for all the pressurized permeation tests conducted in this study.

Figure R8. Mechanical properties of AOPIM-1 adsorption membranes under different treatment conditions.

The relevant discussion has been added in the revised manuscript (page 6, line 21-22; page 7, line 1-2): “All the membranes in various pH condition demonstrate sufficient mechanical strength for pressurized permeation tests (Figure S6), while the mechanical strength appears lower in acid condition, likely due to decreased interchain H-bond interaction.”

Reviewer’s comment: 2) The adsorption mechanisms need in-depth study and analysis. It is known that for these microporous materials with charges, the adsorption of dye molecules may be dominated by electrostatic adsorption on the surface of membranes instead of purely adsorption in the micropores. The schematic diagram shown in Fig. 1c is likely incorrect. Do you have direct evidence to claim that all the dye molecules are adsorbed in the micropores?

Response: It should be clarified in the first place that we did not claim that all the dye molecules are adsorbed in the micropores of the AOPIM-1 membrane. And we agree that the chargeability of the membrane could unavoidably causes adsorption of dye molecules on the membrane surface. But we also have justifications and multiple evidences to support the adsorption of dye molecules in the micropores of the polymer. Firstly, as an amidoxime-modified polymer with intrinsic porosity, AOPIM-1 has high specific surface area and interconnected micropores that provides volume for dye molecule adsorption, and the amidoxime groups further promote adsorption process via electrostatic interactions. The static adsorption experiment performed in this work and previously reported data (B. Satilmis, et al *J Polym Environ.* **28**, 995-1009 (2020); Zhang, C. et al *Chem. Eng. Res. Des.* **109**, 76-85 (2016); Xu, et al, *J. Colloid Interface Sci.* **607**, 890-899 (2022)) confirm the high bulky-adsorption capacity ($490\sim 760\text{ mg g}^{-1}$) of AOPIM-1 through this mechanism. Moreover, we have measured the effective separation pore size of the AOPIM-1 membranes fabricated and included the result in the revised manuscript (Figure S9), which are around 200 nm. This result reveals that the membranes exhibit a hierarchical pore feature, containing macropores generated from the membrane formation process, and intrinsic micropores existing within the polymer matrix. As dye molecules possess sizes in the nanometer range, they are more likely to penetrate into the membrane and get adsorbed into the polymer matrix instead of only remaining on the membrane surface. In addition, we find that the

Brunauer-Emmet-Teller (BET) surface area of the AOPIM-1 adsorptive membrane is reduced from $552 \text{ m}^2 \text{ g}^{-1}$ to $415 \text{ m}^2 \text{ g}^{-1}$ after the adsorption test, and it can be easily regenerated to the original level after desorption (Figure 4b). And it can be seen in Figure 4c that the pore size distribution (PSD) of AOPIM-1 is also restored to the original level after cleaning. Specifically, the amount of 0.5-0.7 nm size micropores reduces the most substantially, indicating that smaller micropores are mostly filled with dye molecules. The variation of membrane's BET and PSD are direct evidences of dye molecules adsorbed in the micropores.

Figure R9. (b) Pore size distributions of pristine, adsorbed and desorbed AOPIM-1 membrane, respectively; (c) Nitrogen absorption-desorption isotherms of pristine, adsorbed and desorbed AOPIM-1, respectively.

Reviewer's comment: 3) The adsorption capacity presented in Figure 3e needs to be explained carefully. Because the molecules are likely adsorbed on the surface of membranes and form a cake layer, the adsorption of these molecules are apparent values instead of intrinsic values. Also the capacity should be linked with the pore volume of the membranes available for adsorption instead of membrane area, because the thickness of these membranes are different. So the adsorptive capacity unit should be changed to g/m^3 instead of g/m^2 .

The authors tried to compare the rejection and flux trade-off with conventional separation membranes based on size and charge. In my opinion, this is not a fair comparison. When the porous polymers are used as adsorbents, the separation performance are dynamically changing, both rejection and water flux are dynamic values. Therefore, the adsorption performance in terms of flux and rejection are not intrinsic properties of the membranes. If the authors intend to compare the performance with other membranes, they should clarify that these membranes are adsorbents, and compare with previously reported membrane adsorbents. For example, MOF membrane adsorbents show water permeance of $20,000 \text{ L m}^{-2} \text{ bar}^{-1} \text{ h}^{-1}$, which is much higher than the water permeance achieved in this study. The MOF membrane adsorbent also showed high rejection at the beginning of filtration tests.

References: High-permeance metal-organic framework-based membrane adsorbent for the removal of dye molecules in aqueous phase. *Environ. Sci.: Nano*, 2017,4, 2205-2214.

Response: We are very grateful that the reviewer has provide in-depth discussion and constructive comments to improve our manuscript. Regarding the adsorption of dye

molecules on the membrane surface, we have provided detailed discussion in the response to Comment #2 from the reviewer. And we agree that the adsorption capacity of the membrane closely relates to the pore volume. In fact, we have demonstrated in the manuscript that the processing capacity of M3 with sponge-like structure is much higher than that of M1 with finger-like structure, revealing that the separation performance of adsorptive membrane is closely related to the whole membrane structure. As recommended by the reviewer, we calculated processing capacity of the AOPIM-1 membranes in the unit of g/m^3 , which is 28, 45, 178, 191 and 219 kg/m^3 for M1, M2, M3, M4, and M5, respectively. Interestingly, the volumetric processing capacity does not only increase sharply when the membrane changes from finger-like structure to sponge-like structure (M1 to M3), but also found to increase with the membrane thickness (M3 to M5). It appears that the membrane processing capacity is closely related to the length of dye transportation route on top of pore volume. The longer and tortuous route would result in higher processing capacity, which is consistent with the property of adsorption materials. The above results and discussions have been included in the revised manuscript (page 10, line 10-19). However, when benchmarking this work with other reported adsorptive membranes, it is difficult to compare the processing capacity with the g/m^3 unit, as the thickness information of related works are often missing. More importantly, separation behavior based on membrane area is the more widely accepted and comparable criteria for the performance evaluation of membranes.

Regarding the review's concern on the performance comparison with other membranes, we agree that the comparison should primarily be conducted among reported adsorptive membranes and we put such comparison in Figure 3e and 3f. Indeed, it is difficult to make comparison with other size/charge selective membranes including nanofiltration and ultrafiltration membranes, but we believe it is of great importance to benchmark our work with membranes that are targeting similar application scenarios for the interest of a wider spectrum of readers. We therefore adopted similar experimental conditions in this study including feed variety, concentration, trans-membrane pressure, temperature, etc. comparable to that of other size/charge selective membranes. The trickiest part is how to collect the effective flux and rejection of adsorptive membranes, as "both rejection and water flux are dynamic values" as mentioned by the reviewer. We hereby clarify that the effective flux and processing capacity of the membrane in this study is calibrated under the criterion of 99% rejection ratio of dye molecule. This criterion is consistent with size/charge selective membranes, and widely accepted by previously published work regarding adsorptive membranes.

Lastly, for the research article referred by the reviewer, although very high permeance has been reported for the MOF membrane, the separation behavior is quite different comparing with our work and other reported adsorptive membranes. According to Figure 8 of the article (show Figure R10 for reference), the dye concentration increases sharply with less than 2 mg rose bengal adsorbed, and the highest membrane rejection recorded in the beginning of filtration tests is around 95% based on reading the first data point in each figure. Although good rose bengal

adsorption capacity of 181.2 mg g^{-1} is reported for the membrane, the dynamic separation behavior of the membrane appears different from our study and other reported adsorptive membranes. Therefore, we feel it inappropriate to include this MOF membrane in the separation performance benchmarking figure (Figure 3f).

Figure R10. Breakthrough curves (Ref. 29) showing dynamic adsorption of rose bengal using the (a) bare α -alumina support and membrane samples: (b) ZIF-8, (c) ZIF-L (no CTAB), (d) ZIF-L (CTAB, 1 \times), (e) ZIF-L (CTAB, 2 \times), and (f) ZIF-L (CTAB, 8 \times).

Reviewer's comment: 4) The multicycle dynamic adsorption and desorption process shown in Figure 4 needs to be carefully explained. Normally, for these adsorbents, as the high-surface area membranes adsorb more molecules and reach equilibrium, the membrane will lose adsorption capacity and the membrane fouling will become dominant. The authors explained that the molecules adsorbed on the surface and inside of the membrane, and likely the water flux decreases. In this case, the phenomenon is membrane fouling. To avoid confusion, the authors should report the original data of the filtration, particularly, the dye concentration of the feed, retentate, and the permeate, so you can estimate the mass balance.

Response: Thanks for reviewer's comment. The multi-cycle adsorptive separation performance stability of the AOPIM-1 adsorptive membrane was performed in a dead-end filtration cell with an effective membrane area of 3.14 cm^2 for 8 cycles. For a typical cycle, the 500 L m^{-2} RHB solution (20 mg/L , $\text{pH} = 10$) was filtrated through the membrane at 0.2 MPa . The original data of every cycle is summarized in the below table and added in the revised supporting information as Table S5. The eluent concentration is provided instead of retentate because the rejected dye is adsorbed on the membrane. It can be seen that permeate concentration slightly increases with cycle number increasing. Thus, the rejection rate slightly decreases with cycle number

increasing due to possible membrane fouling in the cycle filtration process, which is in accordance with the reviewer's comment. However, the membrane fouling could be relieved by longer time desorption treatment as confirmed by the decreased permeate concentration of cycle 8.

Table S5. Concentration of the feed, permeate and eluent (about 20 ml) in each cycle.

Cycle	1	2	3	4	5	6	7	8
Feed (ppm)	20	20	20	20	20	20	20	20
Permeate (ppm)	0.56	0.63	0.52	0.77	0.85	1.18	1.40	0.99
Eluent (ppm)	153.64	151.71	151.45	149.4	150.58	145.14	157.73	--

Figure R11. Processing capacity of AOPIM-1 membranes (coagulation bath composition: H₂O:EtOH = 50:50).

Reviewer's comment: 5) It is recommended to compare the adsorption separation performance with control experiments using AO-PIM powders. For example, you can prepare AO-PIM particles with hierarchical porosity and load the particles into a column and study the adsorption and separation performance.

Response: Thanks for reviewer's suggestion. When AOPIM-1 is made into particles or fibers, the specific surface area might be even larger than that of AOPIM-1 membranes and the adsorption capacity would be optimized. This has been verified by the reported works. Xu et al (Journal of Colloid and Interface Science 607 (2022) 890–899) prepared AOPIM-1/Alg composite beads for cationic dyes adsorption from aqueous solution, achieving high adsorption ability and outstanding regeneration ability. Bekir et al (Applied Surface Science 467-468 (2019) 648-657) has prepared AOPIM-1 fiber ultrafine fibers for rapid removal of uranyl ions from water. However, our work emphasizes the fabrication of an adsorptive membrane with microporous polymer to make the full use of the advantages of AOPIM-1 including high surface area and good solution processability, while simultaneously utilize the size sieving effect of the membrane to achieve highly efficient separation of complex systems such as API feed streams. Despite of the high adsorption capacity, adsorbents in the form of packing columns lack such sieving feature, and are not suitable to be

compared with adsorptive membranes about the separation performance in these application scenarios.

Reviewer's comment: 6) The authors need to change some of the expressions, such as extremely high adsorption capacity, unprecedented capacity.

Response: According to the reviewer's suggestion, we have carefully revised the manuscript for more appropriate expressions, such as changing "extremely high adsorption capacity" to "very high adsorption capacity" and "unprecedented capacity" to "high capacity".

Reviewer's comment: 7) The authors are recommended to perform a deeper literature search, and include relevant reports in the references, for example, recent report on preparation of hierarchical AO-PIM membranes by NIPS method. *Nature Sustainability*, 5, 71–80 (2022).

Response: We appreciate the reviewer's suggestion. A thorough literature search has been conducted accordingly and relevant reports including "Nature Sustainability. 5, 71-80 (2021)" have been included in the references.

Reference

Ref. 20. Wang, H., Zhao, S., Liu, Y., Yao, R., Wang, X., Cao, Y., Ma, D., Zou, M., Cao, A., Feng, X. & Wang, B. Membrane adsorbents with ultrahigh metal-organic framework loading for high flux separations. *Nat. Commun.* **10**, 4204 (2019).

Ref. 39. Yang, L., Xiao, H., Qian, Y., Zhao, X., Kong, X.Y., Liu, P., Xin, W., Fu, L., Jiang, L., & Wen, L. Bioinspired hierarchical porous membrane for efficient uranium extraction from seawater. *Nat. Sustain.* **5**, 71-80 (2021).

Ref. 40. Satilmis, B. Amidoxime Modified Polymers of Intrinsic Microporosity (PIM-1); A Versatile Adsorbent for Efficient Removal of Charged Dyes; Equilibrium, Kinetic and Thermodynamic Studies. *J. Polym. Environ.* **28**, 995-1009 (2020).

Reviewer's comment: 8) To some extent, the AO-PIM membrane reported by the authors are still ultrafiltration membranes. It will be good to compare with recent development of ultrafiltration membranes in the literature. Some recent studies on ultrafiltration membranes also show very high water flux and high rejection towards dye molecules. For example, recent work published in *Separation and Purification Technology*, 2022, 283, 120163, showed, a pure water flux of 110.4 LMH with 99.2% rejection to dye molecules (Congo red).

Response: Thanks for reviewer's important advice. The recent studies on ultrafiltration membranes have been incorporated in Figure 3f as data point for comparison and added in the references. The revised Figure and new references are shown in the below text.

Figure R12. Comparison of AOPIM-1 membranes with reported membranes with respect to their permeation flux and dye rejection/removal efficiency.

Reference

Ref. 53. Zhou, J.Y., Yin, M-J., Wang, Z-P., Shen, Y., Wang, N., Qin, Z., & An, Q-F. Tailoring of polysulfate/polyvinylpyrrolidone membrane structure via NIPS coupled physical aging technique for high-performance dye/salt separation. *Sep. Purif. Technol.* **283**, 120163 (2022).

Ref. 54. Liu, C., Mao, H., Zheng, J., & Zhang, S. Tight ultrafiltration membrane: Preparation and characterization of thermally resistant carboxylated cardo poly (arylene ether ketone)s (PAEKCOOH) tight ultrafiltration membrane for dye removal, *J. Membr. Sci.* **530**, 1-10 (2017).

Ref. 55. Hu, M., Cui, Z., Yang, S., Li, J., Shi, W., Zhang, W., Matindi, C., He, B., Fang, K., & Li, J. Pregelation of sulfonated polysulfone and water for tailoring the morphology and properties of polyethersulfone ultrafiltration membranes for dye/salt selective separation, *J. Membr. Sci.* **618**, 118746 (2021).

Ref. 56. Liu, Y., Wang, J., Wang, Y., Zhu, H., Xu, X., Liu, T., & Hu, Y. High-flux robust PSf-b-PEG nanofiltration membrane for the precise separation of dyes and salts, *Chem. Eng. J.* **405**, 127051 (2021).

Reviewer's comment: 9) Figure S11, it would be better to provide more analysis of the dye molecules, for example, molecular volume and size.

Response: Thanks for reviewer's kind suggestion. We have added the molecule model and molecule size including 3D size, arithmetic mean radius, and geometric mean radius of dye molecules in the Figure S14 of the revised supporting information.

Figure R13. Molecular formula and model size (Å) of different dyes referred in this work.

Reviewers' Comments:

Reviewer #2:

Remarks to the Author:

The author has revised the manuscript based on my comments. The manuscript can be accepted now.

Reviewer #3:

Remarks to the Author:

The authors have made efforts and addressed most of the reviewers' concerns. They have performed additional molecular dynamics simulation and carried out additional experiments to address most of reviewer's comments. These additional results certainly improved the quality of the work and advance the understanding of these new regenerable adsorptive membranes. However, there are still some minor concerns about the explanation of adsorption mechanisms and the presentation of performance data, which can be misleading.

The adsorption mechanism needs to be carefully evaluated. In response to Reviewer 2: the authors explained that "the dye molecules sieving should be mainly attributed to electrostatic interactions". However, in their response to this reviewer, the authors also suggest the adsorption of dye molecules in the micropores of the polymer also needs to be considered. I suggest the authors provide a more accurate description of the adsorption mechanism because the adsorption might involve both electrostatic interactions and adsorption in the micropores, and non-electrostatic interactions could not be fully ruled out. The adsorption also depends on the size and charge of the molecules, and the degree of swelling of membranes in protonated and deprotonated conditions.

I still feel that it is inappropriate to evaluate the membrane adsorption performance simply by comparing water flux and rejection with those of nanofiltration and ultrafiltration membranes. The plot of the initial high rejection value versus water flux is misleading. Rejection is a dynamic changing value, the dynamic adsorption capacity of the membrane (mg g^{-1}) is probably more reasonable, but this value also depends on the porosity of the membranes. If the authors insist to present the plot and use the rejections and value of adsorption capacity (g m^{-2}), then they should clearly state how the values are determined in the figure captions (Fig. 3) and in the main text.

It is appreciated that the authors calculated the volumetric processing capacity in the unit of g/m^3 . According to their explanation, the adsorption capacity is related to the pore structures (sponge or finger-like), and the membrane thickness. In this case, the adsorption is very likely not uniform in the membrane. The molecules would accumulate on the membrane surface region instead of penetrating through the membrane. When the top surface region is saturated and then the membrane rejection starts to decrease. Did the authors observe a cake layer on the membrane surface? This can be easily confirmed by SEM imaging. Understanding the sites and distribution of adsorption of dye molecules in the porous membranes would be highly useful. The authors only provided adsorption data. If the authors could provide some direct experimental evidence of the membrane characterization, for example, the distribution of dye molecules in the membranes, the results will significantly improve the quality of this work. For example, you can dry the dye-saturated membranes by freeze-drying (so you can keep the porous structure of the membranes without collapse), and performing surface and cross-sectional SEM imaging and EDX mapping, especially if the dye molecules contain elements that can be determined by EDX mapping (e.g. sulfur). These results would provide direct evidence of the adsorption and distribution of molecules in the membranes. Actually, this is widely used for studying mechanisms of membrane fouling in microfiltration and ultrafiltration membranes.

Overall, the authors have performed nice work on regenerable adsorptive membranes.

Comment 1: The authors have made efforts and addressed most of the reviewers' concerns. They have performed additional molecular dynamics simulation and carried out additional experiments to address most of reviewer's comments. These additional results certainly improved the quality of the work and advance the understanding of these new regenerable adsorptive membranes. However, there are still some minor concerns about the explanation of adsorption mechanisms and the presentation of performance data, which can be misleading.

The adsorption mechanism needs to be carefully evaluated. In response to Reviewer 2: the authors explained that "the dye molecules sieving should be mainly attributed to electrostatic interactions". However, in their response to this reviewer, the authors also suggest the adsorption of dye molecules in the micropores of the polymer also needs to be considered. I suggest the authors provide a more accurate description of the adsorption mechanism because the adsorption might involve both electrostatic interactions and adsorption in the micropores, and non-electrostatic interactions could not be fully ruled out. The adsorption also depends on the size and charge of the molecules, and the degree of swelling of membranes in protonated and deprotonated conditions.

Response: We are very grateful for the reviewer's suggestion on how to improve our explanation of the membrane adsorption mechanism. It appears that our previous response has caused some ambiguities, and the role of electrostatic interactions and micropores on the membrane adsorption needs further discussion and clarification.

Basically, we believe that the adsorption of dye molecules is mainly attributed to electrostatic interactions, which specifically refers to the electrostatic attraction between charged amidoxime groups of AOPIM-1 and oppositely charged dye molecules. As shown in Figure R1, AOPIM-1 favors the capture of negatively charged MO molecules under acidic conditions while capturing the oppositely charged MB under alkaline conditions with the equilibrium adsorption capacity higher than 450 mg g⁻¹. And the capacity falls below 100 mg g⁻¹ under neutral pH conditions where the amidoxime group possess minimal chargeability, which reveals that the pH-tunable affinity sites make major contribution to the adsorption capacity. Therefore, the dye molecules separation by AOPIM-1 membrane should be mainly attributed to electrostatic interactions between charged amidoxime groups and oppositely charged dye molecules, which is also supported by the molecular dynamics simulation result provided in the previous revision.

However, we also agree with the reviewer that electrostatic attraction is not the only reason for the high capacity dynamic membrane adsorption behavior observed. As a control experiment, a negatively charged polyethersulfone ultrafiltration membrane with similar pore size and water flux was selected to filtrate feed solution containing positively charged MB dye molecules. It is found that almost all dye molecules pass through the membrane without noticeable retention (Figure R2), which is common for conventional ultrafiltration membranes that rejects large molecules and particulate matter but cannot accurately separate small organic molecules. Obviously, polymer materials without intrinsic microporosity (polyethersulfone in this case) lack necessary specific surface area and affinity sites to

allow the electrostatic attraction between functional groups and target molecules to occur. In fact, Figure R1 also reveals that a minor portion of the adsorption capacity relies on non-electrostatic interactions, as the capacity does not completely drop to zero under neutral pH condition. This remaining capacity is also a demonstration of what micropores can bring to the membrane adsorption without the help from electrostatic interactions.

Based on the above results and discussions, we could conclude that the electrostatic attraction and the microporous feature of AOPIM-1 co-govern the dynamic adsorption performance of the membrane. The charged molecules electrostatically interact with the charged amidoxime groups inside the AOPIM-1 polymer matrix, and the microporous structure of AOPIM-1 provides abundant adsorption sites for the high capacity adsorption of charged molecules during dynamic membrane separations.

The above discussion about the membrane adsorption mechanism has been included in the revised manuscript as follows: “As shown in Figure 2b, AOPIM-1 favors the capture of negatively charged MO molecules under acidic conditions while capturing the oppositely charged MB under alkaline conditions with the equilibrium adsorption capacity higher than 450 mg g⁻¹. And the capacity falls below 100 mg g⁻¹ under neutral pH conditions where the amidoxime group possess minimal chargeability, which reveals that the pH-tunable affinity sites make major contribution to the adsorption capacity. Therefore, the dye molecules separation by AOPIM-1 membrane should be mainly attributed to electrostatic interactions between charged amidoxime groups and oppositely charged dye molecules.” (page 7, line 15-22; page 8, line 1-2) and “As a control experiment, a negatively charged polyethersulfone ultrafiltration membrane with similar pore size and water flux is selected to filtrate feed solution containing positively charged MB dye molecules. It is found that almost all dye molecules pass through the membrane without noticeable retention (Figure S17), which is common for conventional ultrafiltration membranes that rejects large molecules and particulate matter (such as proteins, suspended solids, bacteria, viruses, and colloids) but cannot accurately separate small organic molecules¹². Obviously, polymer materials without intrinsic microporosity (polyethersulfone in this case) lack necessary specific surface area and affinity sites to allow the electrostatic attraction between functional groups and target molecules to occur. Based on the above results and discussions, it could be concluded that the electrostatic attraction and the microporous feature of AOPIM-1 membrane co-govern the dynamic adsorption performance of the membrane. The charged molecules electrostatically interact with the charged amidoxime groups inside the AOPIM-1 polymer matrix, and the microporous structure of AOPIM-1 provides abundant adsorption sites for the high capacity adsorption of charged molecules during dynamic membrane separations.” (page 13, line 21-22; page 14, line 1-16).

Figure R1. Equilibrium adsorption capacity (q_e) of AOPIM-1 for MO and MB dye molecules at different pH conditions.

Figure R2. Separation performance of polyethersulfone ultrafiltration membrane and AOPIM-1 membrane against the MB dye feed solution.

Comment 2: I still feel that it is inappropriate to evaluate the membrane adsorption performance simply by comparing water flux and rejection with those of nanofiltration and ultrafiltration membranes. The plot of the initial high rejection value versus water flux is misleading. Rejection is a dynamic changing value, the dynamic adsorption capacity of the membrane (mg g⁻¹) is probably more reasonable, but this value also depends on the porosity of the membranes. If the authors insist to present the plot and use the rejections and value of adsorption capacity (g m⁻²), then they should clearly state how the values are determined in the figure captions (Fig. 3) and in the main text.

Response: We appreciate that the reviewer has made very sincere comments regarding the data presentation and bench-marking. As what we have clarified in the previous response, the effective flux and processing capacity of the membrane in this study is calibrated under the criterion of 99% rejection ratio of dye molecule. This criterion is widely accepted by previously published work regarding adsorptive

membranes. And we believe that the comparison with other size/charge selective membranes is of great importance to benchmark our work with membranes targeting similar application scenarios. To make comparison, we therefore adopted similar experimental conditions in this study including feed variety, concentration, trans-membrane pressure, temperature, etc. comparable to that of other size/charge selective membranes.

According to reviewer's suggestion, the statement "It should be mentioned that the effective processing capacity of membranes is calibrated under the criterion of 99% rejection ratio of Rhodamine B (RHB) (20 ppm)" has been clarified in the revised manuscript (page 9, line 18-20). The statement "The effective flux and processing capacity of the membrane in this study is calibrated under the criterion of 99% rejection ratio of dye molecule." has been added in the figure captions (Fig. 3).

Comment 3: It is appreciated that the authors calculated the volumetric processing capacity in the unit of g/m³. According to their explanation, the adsorption capacity is related to the pore structures (sponge or finger-like), and the membrane thickness. In this case, the adsorption is very likely not uniform in the membrane. The molecules would accumulate on the membrane surface region instead of penetrating through the membrane. When the top surface region is saturated and then the membrane rejection starts to decrease. Did the authors observe a cake layer on the membrane surface? This can be easily confirmed by SEM imaging. Understanding the sites and distribution of adsorption of dye molecules in the porous membranes would be highly useful. The authors only provided adsorption data. If the authors could provide some direct experimental evidence of the membrane characterization, for example, the distribution of dye molecules in the membranes, the results will significantly improve the quality of this work. For example, you can dry the dye-saturated membranes by freeze-drying (so you can keep the porous structure of the membranes without collapse), and performing surface and cross-sectional SEM imaging and EDX mapping, especially if the dye molecules contain elements that can be determined by EDX mapping (e.g. sulfur). These results would provide direct evidence of the adsorption and distribution of molecules in the membranes. Actually, this is widely used for studying mechanisms of membrane fouling in microfiltration and ultrafiltration membranes. Overall, the authors have performed nice work on regenerable adsorptive membranes.

Response: According to the reviewer's suggestion, we have conducted relevant SEM characterization of "saturated" AOPIM-1 membrane - the membrane at the point where the dye rejection falls below 99% during the dynamic adsorption experiment. Methylene blue (MB) is selected as adsorbate because MB contains sulfur elements that can be determined by EDX mapping. The membrane is dried via freeze-drying to avoid possible structural feature collapse. The obtained membrane is characterized by SEM imaging and EDX mapping. The surface and cross-sectional SEM and EDX mapping images are shown in Figure R3. There is no obvious cake-layer found in the membrane surface. Instead, it could be clearly seen that the sulfur element is widely distributed across the whole cross-section of the membrane, especially among the

polymer-rich regions. This result provides direct evidence of the adsorption and distribution of molecules in the AOPIM-1 membrane. It also demonstrates directly that the dye molecules are adsorbed in the polymer micropores across the entire membrane until saturation, and then penetrates through the membrane afterwards.

We believe this direct observation greatly advances the understanding of the dynamic adsorption behavior of AOPIM-1 membranes, and would like to show our gratitude again for the constructive comments from the reviewer to improve our manuscript.

The SEM characterization and discussion has been added in the Supporting Information as Figure S14, and in the revised manuscript (page 10, line 20-22; page 11, line 1-14): “To provide direct evidence of the adsorption and distribution of molecules in the membranes, SEM characterization of “saturated” AOPIM-1 membrane (the membrane at the point where the dye rejection falls below 99% during the dynamic adsorption experiment) is conducted. Methylene blue (MB) is selected as adsorbate because MB contains sulfur elements that can be determined by EDX mapping. The membrane is dried via freeze-drying to avoid possible structural feature collapse. The obtained membrane is characterized by SEM imaging and EDX mapping. The surface and cross-sectional SEM and EDX mapping images are shown in Figure S14. There is no obvious cake-layer found in the membrane surface. Instead, it could be clearly seen that the sulfur element is widely distributed across the whole cross-section of the membrane, especially among the polymer-rich regions. This result provides direct evidence of the adsorption and distribution of molecules in the AOPIM-1 membrane. It also demonstrates directly that the dye molecules are adsorbed in the polymer micropores across the entire membrane until saturation, and then penetrates through the membrane afterwards.”.

Figure R3. The surface and cross-sectional (a,c,e) SEM images and sulfur element distribution (b,d,f) in EDX mapping images of AOPIM-1 membrane after the dynamic adsorption experiment.

Reviewers' Comments:

Reviewer #3:

Remarks to the Author:

The team have made addressed further concerns raised by this reviewer and performed additional experiments on SEM analysis, which improved the quality of the work. The revised manuscript is suitable for publication.

One last suggestion: some of the hype claims in the title and abstract need to be changed: such as Unprecedented Processing Capacity can be changed to high adsorption capacity.